# *In vivo* optochemical control of cell contractility at single-cell resolution

Deqing Kong[1,2,3,*] iD, Zhiyi Lv[1], Matthias Häring[3,4,5,6,7], Benjamin Lin[8], Fred Wolf[3,4,5,6,7,†] & Jörg Großhans[1,2,†] iD

## Abstract

The spatial and temporal dynamics of cell contractility plays a key role in tissue morphogenesis, wound healing, and cancer invasion. Here, we report a simple optochemical method to induce cell contractions *in vivo* during *Drosophila* morphogenesis at single-cell resolution. We employed the photolabile $Ca^{2+}$ chelator *o*-nitrophenyl EGTA to induce bursts of intracellular free $Ca^{2+}$ by laser photolysis in the epithelial tissue. $Ca^{2+}$ bursts appear within seconds and are restricted to individual target cells. Cell contraction reliably followed within a minute, causing an approximately 50% drop in the cross-sectional area. Increased $Ca^{2+}$ levels are reversible, and the target cells further participated in tissue morphogenesis. Depending on Rho kinase (ROCK) activity but not RhoGEF2, cell contractions are paralleled with non-muscle myosin II accumulation in the apico-medial cortex, indicating that $Ca^{2+}$ bursts trigger non-muscle myosin II activation. Our approach can be, in principle, adapted to many experimental systems and species, as no specific genetic elements are required.

**Keywords** actomyosin; $Ca^{2+}$ uncaging; cell contractility; morphogenesis; optochemical

**Subject Categories** Cell Adhesion, Polarity & Cytoskeleton; Methods & Resources

## Introduction

Contractility underlies manifold processes in cell and tissue morphogenesis, including cell migration, cell shape changes, or junction collapse [1–4]. In epithelial tissues, cell contractions impact neighboring cells by exerting forces on adherens junctions. This mechanical linkage may elicit specific responses and could thus positively or negatively affect contractility and cytoskeletal organization in neighboring cells, i.e., mediate non-autonomous mechanical behaviors [5]. Within a tissue, cellular contraction and cell–cell interactions based on such force transduction can contribute to emergent tissue behavior, such as the formation of folds and furrows. The function of mutual cell–cell interactions, however, is difficult to study by classical genetic approaches. What is needed are methods for acute noninvasive interventions with high temporal and spatial resolution, ideally on the scale of seconds and of single cells.

For controlling cell contractility, optogenetic approaches have recently been developed. Cell contractility can be inhibited by optically induced membrane recruitment of $PI(4,5)P_2$ leading to interference with phosphoinositol metabolism and subsequent suppression of cortical actin polymerization [6]. Optical activation of contractility has been achieved by light-induced activation of the Rho-ROCK (Rho kinase) pathway, which controls myosin II-based contractility [7,8]. While functionally effective, such optogenetic methods require multiple transgenes driving the expression of modified proteins such as light-sensitive dimerization domains, which restrict the application to genetically tractable organisms. In addition, chromophores used in optogenetic effectors are activated by light in the visible spectrum, which limits the choice of labels and reporters for concurrent cell imaging.

Optochemical methods represent an alternative to genetically encoded sensor and effector proteins [9]. Intracellular calcium ions ($Ca^{2+}$) are known to be an important regulator of contractility in many cell types. $Ca^{2+}$ plays a central role not only in muscle contraction, but also in cultured epithelial cells [10], in amnioserosa cells during dorsal closure [11], during neural tube closure [12,13], and in the folding morphogenesis of the neural plate [14]. In *Drosophila* oogenesis, tissue-wide increase in intracellular $Ca^{2+}$ activates myosin II and impairs egg chamber elongation [15]. In *Xenopus*, a transient increase in $Ca^{2+}$ concentration induces apical constriction

1  Institute for Developmental Biochemistry, Georg-August-Universität Göttingen, Göttingen, Germany
2  Faculty of Biology, Philipps-Universität Marburg, Marburg, Germany
3  Bernstein Center for Computational Neuroscience, Göttingen, Germany
4  Max Planck Institute for Dynamics and Self-Organization, Göttingen, Germany
5  Institute for Nonlinear Dynamics, Georg-August-Universität Göttingen, Göttingen, Germany
6  Campus Institute for Dynamics of Biological Networks, Göttingen, Germany
7  Max Planck Institute for Experimental Medicine, Göttingen, Germany
8  Department of Cell Biology, HHMI and Kimmel Center for Biology and Medicine of the Skirball Institute, New York University School of Medicine, New York, NY, USA
   *Corresponding author. Tel: +49 551 39 68273; E-mail: deqing.kong@med.uni-goettingen.de
   †These author contributed equally to this work as co-senior authors

in cells of the neural tube [16]. Although the detailed mechanism of $Ca^{2+}$-induced contraction in non-muscle cells remains to be resolved, it conceivably offers a simple and temporally precise way to interfere with and control contractile activity. In neuroscience, optochemical methods for the release of intracellular $Ca^{2+}$ have been well established and widely employed [17,18]. Here, we report an optochemical method to control epithelial cell contractility via $Ca^{2+}$-mediated light activation of myosin (CaLM) on the scale of seconds and at single-cell resolution during tissue morphogenesis in *Drosophila* embryos. Optochemical control of contractility by $Ca^{2+}$ uncaging has minimal spectral overlap with fluorescent protein reporters and optogenetic activators. Our results provide evidence for a ROCK-dependent effect of increased intracellular $Ca^{2+}$ on activating non-muscle myosin II and its recruitment to the actomyosin cortex.

## Results

### Uncaging induces a rapid $Ca^{2+}$ burst in epithelial cells in *Drosophila* embryos

Photolysis of the $Ca^{2+}$ chelator *o*-nitrophenyl EGTA (NP-EGTA) [19] (Fig 1A) is widely used in neurobiology for the modulation of intracellular $Ca^{2+}$ concentration [18,20]. Here, we employed the membrane-permeant acetoxymethyl (AM) ester derivative, which complexes $Ca^{2+}$ once the AM moiety is cleaved off by intracellular esterase. The *o*-nitrophenyl EGTA-$Ca^{2+}$ complex cannot get out again because the AM moiety has been cleaved off by intracellular esterase. Following microinjection into staged embryos, uncaging was induced in the focal volume with a diameter of 2–3 μm and thus an area of 5 μm² of a pulsed 355-nm laser beam (Fig 1B). To allow for concomitant uncaging and imaging, we used a setup, in which the light paths of the UV laser for uncaging and the excitation laser for confocal imaging in the visible spectrum were controlled independently. We conducted experiments in the lateral epidermis of *Drosophila* embryos during germband extension (stage 7). The epidermis during this stage constitutes a columnar epithelium with a cell diameter in the range of about 8 μm and cell height of about 25 μm (Fig 2A).

We recorded changes in intracellular $Ca^{2+}$ concentration induced by uncaging using a genetically encoded $Ca^{2+}$ sensor protein, GCaMP6s. Embryos expressing a membrane-bound, myristoylated variant of GCaMP6s [21] were injected with NP-EGTA-AM and subjected to uncaging. We observed a transient increase in GCaMP6 fluorescence within a second specifically in cells targeted by a UV light pulse (Fig 1C, Movie EV1). Quantification of GCaMP fluorescence ($\Delta F/F_0$) showed a fourfold increase within 2-s. Afterward, GCaMP6s fluorescence gradually decreased to near initial levels within a few minutes (Fig 1E). As GCaMP6s has a decay time constant in the range of seconds, this indicates that $Ca^{2+}$ clearance and extrusion mechanisms in the epithelial cells operate on an effective time scale of minutes. We did not detect an increase in GCaMP6s fluorescence after UV exposure in control embryos injected with buffer only (Fig 1D and E).

The increase in the $Ca^{2+}$ sensor signal was restricted to the individual target cell (Fig 1C, Movie EV1). The $Ca^{2+}$ sensor signal in the next and next–next neighbors of the target cell was temporally

constant and comparable to control embryos (Fig 1F). In summary, our experiments show that $Ca^{2+}$ uncaging with single-cell precision can be conducted in epithelial tissue in *Drosophila* embryos. Uncaging leads to a reversible, second-scale increase in intracellular $Ca^{2+}$ concentration that is restored by cell-intrinsic mechanisms on a minute scale. The magnitude of the $Ca^{2+}$ increase was similar to what was previously reported for neuronal cells [22].

### $Ca^{2+}$ bursts induce cell contraction

We next investigated the consequence of $Ca^{2+}$ bursts on cell shape. We conducted uncaging in embryos expressing E-Cad-GFP, which labels adherens junctions near the apical surface of the epithelium (Fig 2A). We detected a contraction of the target cell in the lateral epidermis to about half of the apical cross-sectional area following uncaging (Figs 2B and EV1A, Movies EV2 and EV3). Target cells in control embryos injected with buffer remained largely unaffected (Fig 2C). Quantification revealed a reduction by half of the cross-sectional area within 1–2 min in the target cell but not in controls (Figs 2D, and EV1A and B). The constriction rate reached the maximum in 0.5 min (Fig 2E). Most cells remained contracted during the following 15 min, whereas a minority of cells reexpanded to the original cross-sectional area (Fig 2F and G). We did not observe that the exposure to UV laser and $Ca^{2+}$ uncaging noticeably affected the further behavior of the target cells and surrounding tissue (Fig 2F and G). We did not observe that target cells were extruded or got lost from epithelial tissue. This behavior indicates that the $Ca^{2+}$ uncaging is compatible with ongoing tissue morphogenesis. We conducted $Ca^{2+}$ uncaging in the head and dorsal region at stage 7 embryos, where these cells do not display apical myosin and do not display obvious changes in cross-sectional area. Cell contraction event was detected in these cells following $Ca^{2+}$ uncaging (Fig EV1C and D).

### Induced cell contraction in a squamous epithelium

Next, we applied $Ca^{2+}$ uncaging to a different tissue in *Drosophila* embryos. The amnioserosa represents a squamous epithelium on the dorsal side of the embryo with cells about 15 μm in diameter and only 3 μm in height (Fig 3A–C). As in the lateral epidermis, we employed E-Cadherin-GFP to label the apical cell outlines (Fig 3B). $Ca^{2+}$ uncaging led to contraction of the target cells but not in the control cells (Fig 3D, Movie EV4). The cells that are from the same recording but not the next-neighboring of target cell were used as control (Fig 3D). Quantification of the apical cross-sectional areas revealed specific uncaging-induced contraction within a minute, and the peak constriction rate was observed about 30 s after uncaging (Fig 3E). The amnioserosa cells are naturally contracting overtime (Fig 3F). We calculated the maximum constriction rate from 12 control cells over 5 min and detected a statistically significant difference when comparing the maxima in the constriction rates between the target and control cells (Fig 3G). We next conducted three uncaging experiments in amnioserosa cells with recording over 30 min. Two cells contracted irreversibly, one cell relaxed after 10 min as in the lateral epidermis (Fig EV2B and C). We did not observe that the exposure to UV laser and $Ca^{2+}$ uncaging noticeably affected the further behavior of the target cells and surrounding

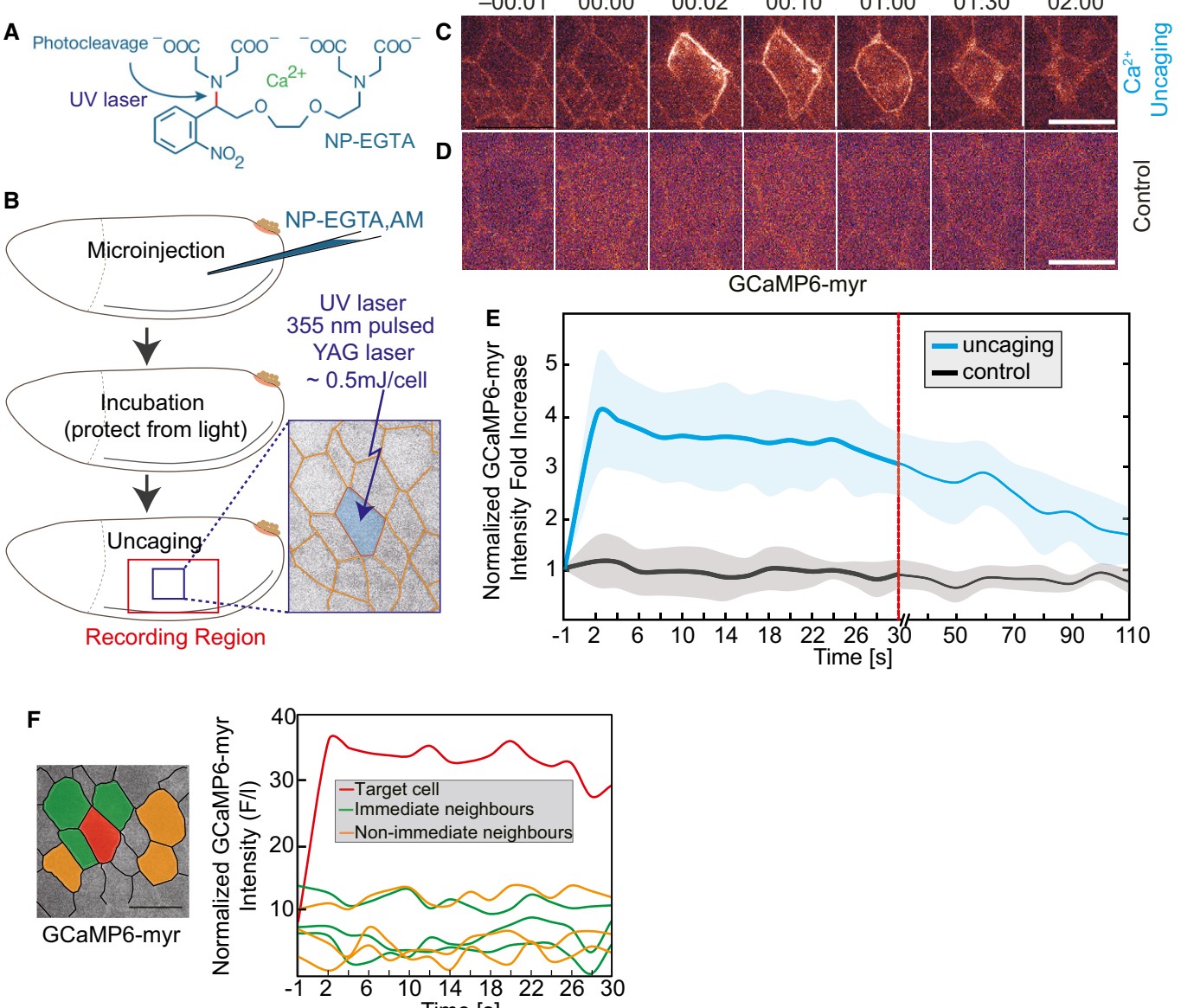

**Figure 1. CaLM induces a rapid increase in intracellular Ca²⁺ concentration in epithelial cells.**

A    Structure of the cage NP-EGTA. UV illumination cleaves the bond in red and releases Ca²⁺.

B    Experimental scheme for Ca²⁺ uncaging in *Drosophila* embryos. NP-EGTA, AM was injected into the staged embryos. Followed by a short incubation, a target cell (blue) was exposed to a UV laser flash.

C, D  Images from time-lapse recording of embryos (stage 7, lateral epidermis) expressing a membrane-bound Ca²⁺ sensor (GCaMP6-myr) and injected with (C) 2 mM NP-EGTA, AM or (D) with buffer (control). Time in min:s.

E    Normalized fluorescence intensity of GCaMP-myr in the target cell. Mean (bold line, six cells in six embryos) with standard deviation of the mean (ribbon band).

F    Normalized fluorescence intensity of GCaMP sensor in target cell (red), three next neighbors (green), and three non-immediate neighbors (orange).

Data information: scale bars: 10 μm in (C, D, F).

tissue. Furthermore, in order to rule out that UV laser induced cell apoptosis during uncaging, we employed a reporter of apoptosis [23,24] in the amnioserosa, where we can demonstrate the functionality of the reporter due to the normal presence of apoptotic cells during dorsal closure (Fig EV2A). We detected reporter signal in apoptotic cells but not in target cells subject to uncaging. In summary, our experiments show that Ca²⁺ uncaging can be

employed as a noninvasive method to induce contractions in selected single cells in different cell types and tissues.

We next ask whether further contraction in the target cell can be generated by repeating the UV pulse in the same cell. We therefore exposed a selected cell in the amnioserosa three times with a UV pulse (0, 2.5, and 5 min). We observed the typical contraction after the first pulse but no further obvious contractions after the second

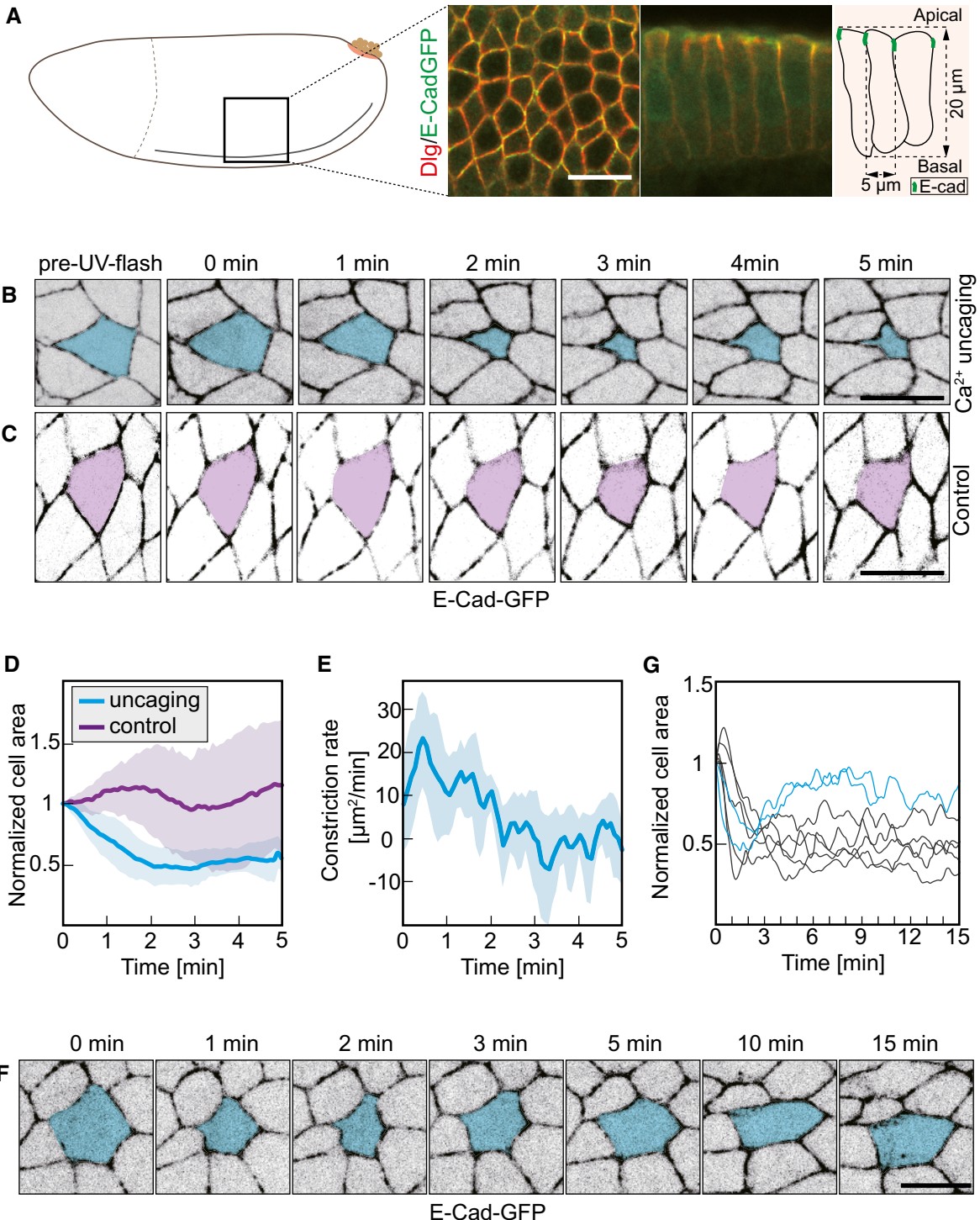

**Figure 2. CaLM triggers apical constriction in a columnar epithelium.**

A       Schematic drawing and morphology of columnar epithelium in the lateral epidermis in stage 7 *Drosophila* embryos.

B, C    Images from a time-lapse recording embryos expressing E-Cad-GFP and injected with (B) 2 mM NP-EGTA, AM or (C) buffer and exposed to the UV laser. Target cells are labeled in blue or purple.

D       Cross-sectional area of target cells over time. Cell areas were normalized to their initial size (the first frame of recording after uncaging). Mean (bold line) with standard deviation of the mean (ribbon band). Uncaging (blue), eight cells in eight embryos. Control (purple), five cells in five embryos.

E       Apical constriction rate over time in target cells in (D) (*n* = 8 cells in eight embryos). Mean (bold line) with standard deviation of the mean (ribbon bands).

F       Images from time-lapse recording showing long-term behavior after CaLM. Target cell is marked in blue.

G       Cross-sectional area of target cells over 15 min after Ca²⁺ uncaging. Cell contraction was reversible in two out of seven target cells (blue lines).

Data information: scale bars: 10 μm in (A, B, C, F).

        

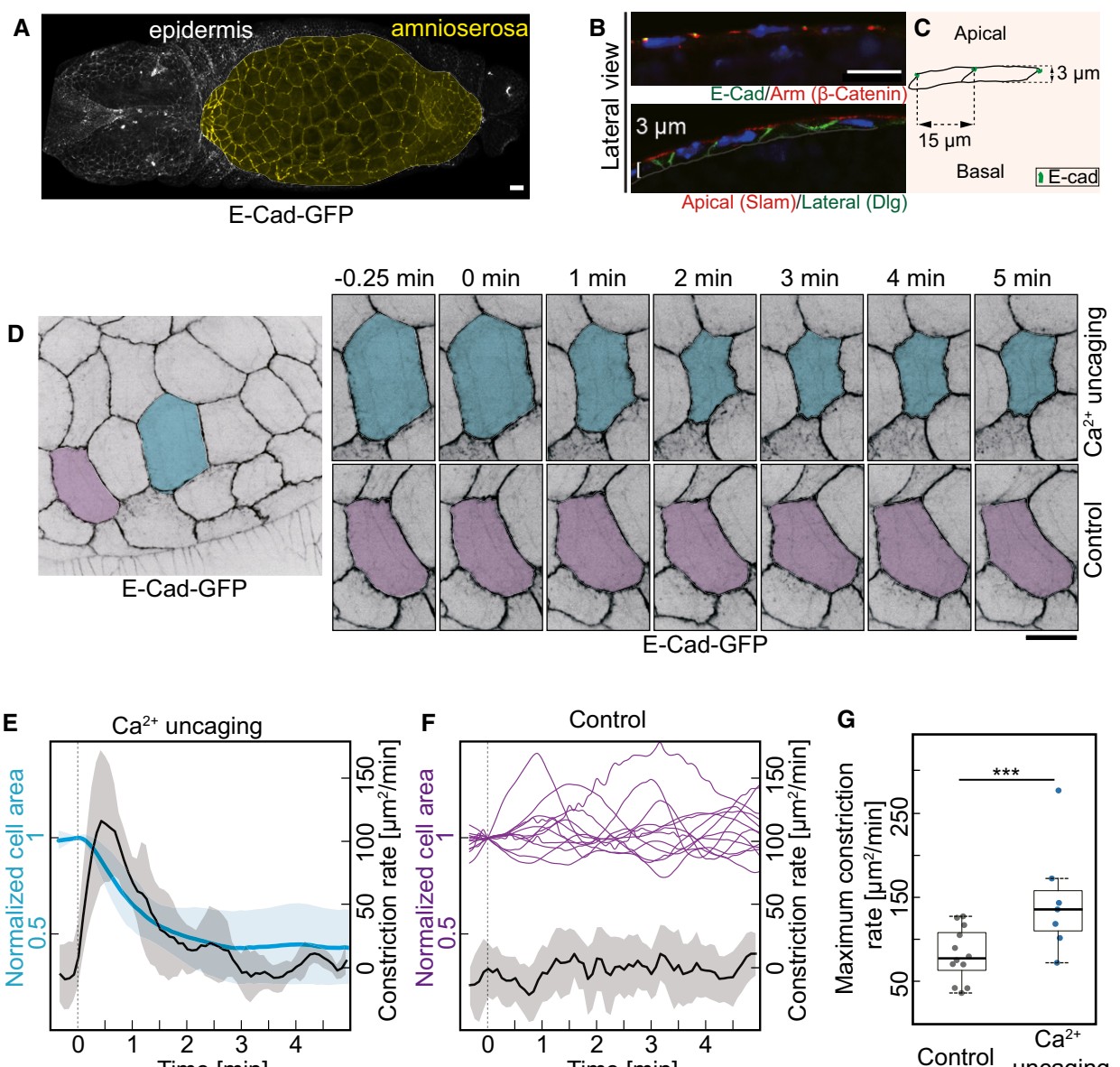

**Figure 3. CaLM triggers apical constriction in a squamous epithelium.**

A–C Amnioserosa (yellow in A) represents a squamous epithelium. Confocal image of *Drosophila* embryo expressing E-Cadherin-GFP. Sagittal sections of amnioserosa cells. Confocal images (B) and schematic drawing (C).

D Images from a time-lapse recording in embryos (stage 14) expressing E-Cad-GFP and injected with 1 mM NP-EGTA, AM. The target cell is highlighted in blue. The control cell (next–next neighbor of target cells) highlighted in purple was not exposed to UV light.

E Cross-sectional area (blue) and apical constriction rate (black) of target cells normalized to initial size (the first frame of recording after uncaging). Mean (bold line) with standard deviation of the mean (ribbon band) (n = 7 cells in seven embryos).

F Cross-sectional area traces (purple) of 12 individual control cells. Mean of apical constriction rate of control cells is indicated with black bold line (n = 12 cells in seven embryos) with standard deviation of the mean (ribbon band).

G Boxplot shows the maximum apical constriction rate from target and control cells. Bold horizontal line, mean. Box, second and third quartile. Black horizontal dash line with whisker, 95% bootstrap confidence intervals. ***P = 0.00004949 (two-tailed unpaired *t*-test).

Data information: Scale bars: 10 μm in (A, B, D).

and third UV pulses (Fig EV3A and B). Next, we induced contraction by uncaging in a row of four cells in the amnioserosa (Fig EV3C). An axial projection after 5 min shows a small groove in the tissue. Importance of this study is that we demonstrate the

induced contraction of a row of cells. Having the method in hand to induce cell contraction in a selected patch of cells will allow us to test the contribution of contraction of morphogenetic movements such as furrow formation and invagination in future experiments.

### Role of myosin II in Ca²⁺-induced cell contraction

Multiple mechanisms are conceivable for $Ca^{2+}$-induced cell contraction. Given their time scale in the minute range, it is unlikely that slow transcriptional or translational processes are involved. It is also unlikely that $Ca^{2+}$ directly activates contraction similar to its role in muscle cells due to the distinct organization of cortical actomyosin and indicated by the substantial time lag between $Ca^{2+}$ increase and cell contraction. $Ca^{2+}$ may activate myosin II, similar to what has been reported for the *Drosophila* egg chamber [15]. Such a specific myosin II activation may be mediated via Rho-ROCK signaling or

via $Ca^{2+}$-dependent protein kinases or phosphatases, such as myosin light-chain kinase (MLK) [25].

As a first step toward identifying the mechanism of $Ca^{2+}$-induced cell contraction, we imaged myosin II dynamics following uncaging in embryos expressing E-Cad-GFP to label cell–cell contacts and sqh-mCherry (spaghetti squash, myosin regulatory light chain). sqh-mCherry fluorescence is a direct indicator of active myosin II mini filaments, which are visible as clusters. Myosin II is found associated with adherens junctions (junctional pool) and at the apical cortex (medial pool), where it is responsible for apical constriction [26]. We focused on the medial pool of myosin II. We observed an

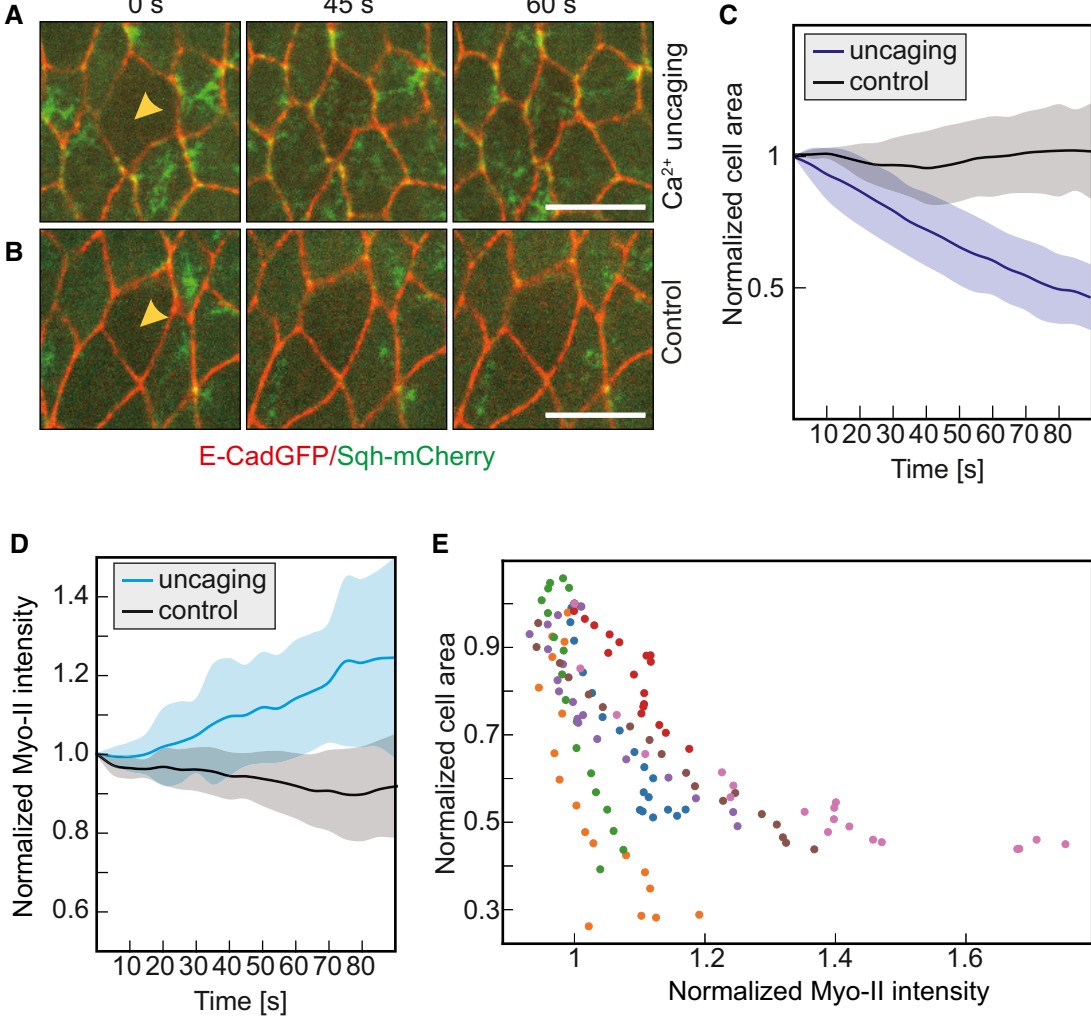

**Figure 4. CaLM induces myosin II.**

A, B   Embryos expressing Sqh-mCherry (green) and E-Cadherin-GFP (red) were injected with 2 mM NP-EGTA, AM (A) or buffer (B). Images from a time-lapse recording in the cells of the lateral epidermis (stage 7) and exposed to the UV laser (yellow arrowheads).

C   Cross-sectional area of target cells and control cells normalized to the initial area (the first frame of recording after uncaging or UV laser illumination). Mean (bold line) with standard deviation of the mean (ribbon band) (*n* = 7 cells in seven embryos).

D   Medio-apical Sqh-mCherry fluorescence in target (blue) and control (black) cells normalized to the initial fluorescence intensities (the first frame of recording after uncaging or UV laser illumination). Mean (bold line) with standard deviation of the mean (ribbon band) (*n* = 7 cells in seven embryos), *P* = 0.013 at 45 s (CE50), *P* = 0.011 at 90 s (two-tailed unpaired *t*-test).

E   Scatter plot of normalized medio-apical myosin II (the first frame of recording after uncaging is normalized to 1) with normalized cross-sectional area (the first frame of recording after uncaging is normalized to 1) in target cells. Different colors indicate the individual cells.

Data information: Scale bars: 10 μm in (A, B).

increase in sqh-mCherry fluorescence after about 0.5–1 min specifically in target cells (Fig 4A and B). Quantification of the medial myosin II revealed specific uncaging induced a 20% increase in

target cells within 1.5 min after uncaging (Fig 3D). However, medial myosin II intensity dropped a bit in the control cells following UV exposure from the embryos injected with buffer without NP-EGTA,

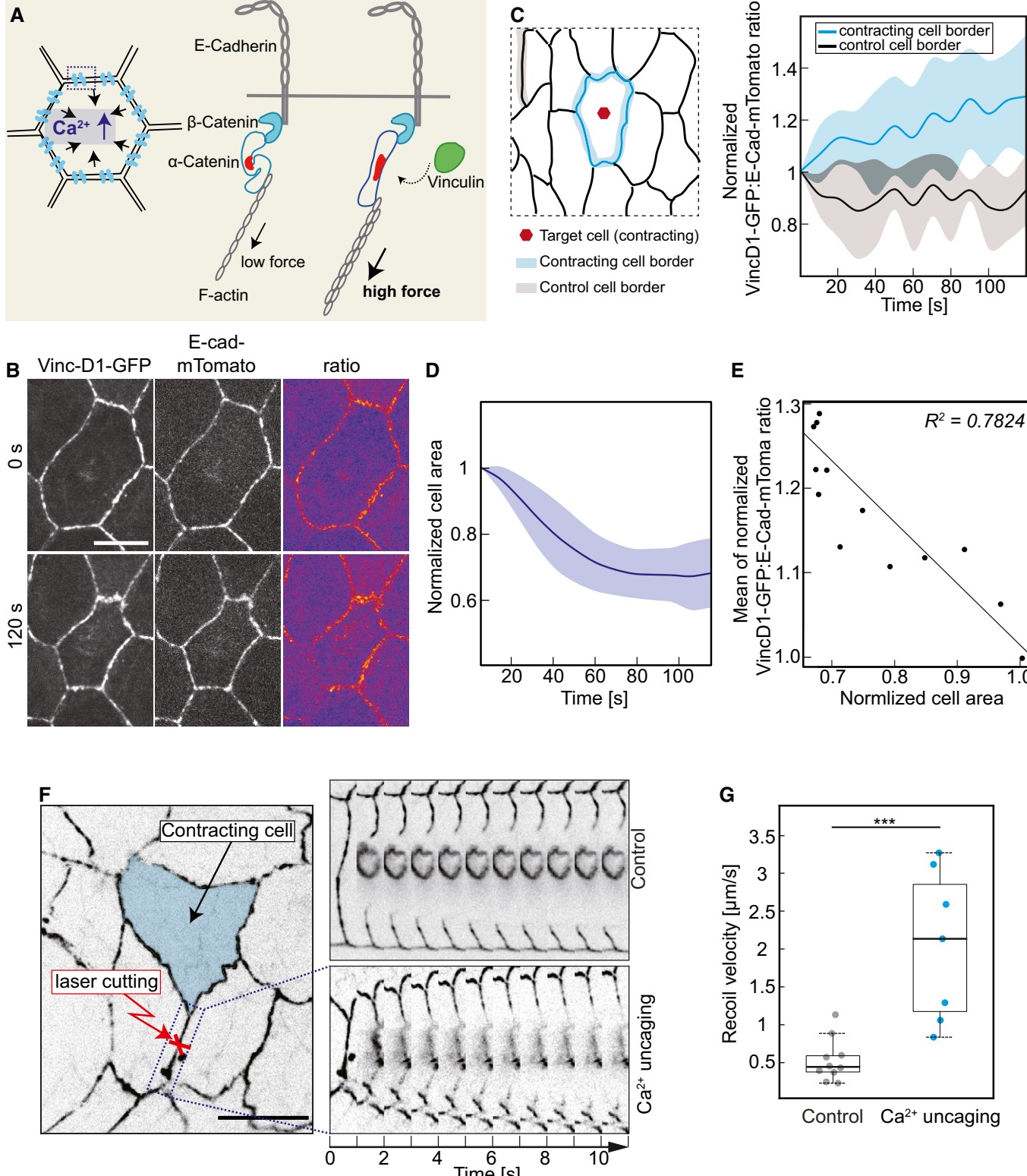

**Figure 5.**

◀

**Figure 5. CaLM induces cortical tension.**

A   Schematic drawing of force-dependent Vinculin association to adherens junctions and principle of the Vinculin reporter.
B   Images from time-lapse recording of an amnioserosa cell after CaLM in embryos (stage 14) expressing E-Cad-mTomato and VinculinD1-GFP.
C   Ratio of VinculinD1-GFP and E-Cadherin-mTomato fluorescence at the junctions of the target contracting cells (blue) and control cells (black) The ratio was normalized to initial ratio (the first frame of recording after uncaging). Mean (bold line) with standard deviation of the mean (ribbon band) (*n* = 6 constricting cells and nine inactive cell borders in six embryos), *P* = 0.033 at 60 s (CE50), *P* = 0.011 at 120 s (two-tailed unpaired *t*-test).
D   Cross-sectional area in target cells normalized to initial size (the first frame of recording after uncaging). Mean (bold line) with standard deviation of the mean (ribbon band) (*n* = 6 cells in six embryos).
E   Scatter plot of normalized area of target cells with the mean of VinculinD1/E-Cadherin ratio at the cell junctions (*n* = 6 cells in six embryos).
F   The schematic of amnioserosa shows the first neighbor junction of CaLM target cells (indicated by red cross). Kymographs show recoil after junction ablation. Control ablations were conducted in the embryos injected with buffer without NP-EGTA, AM, and the junctions were selected randomly.
G   Boxplot shows the initial recoil velocity after laser ablation. Bold horizontal line, mean. Box, second and third quartile. Black horizontal dash line with whisker, 95% bootstrap confidence intervals. ***$P$ = 0.00035151 (two-tailed unpaired *t*-test). Dots indicate the individual recoil velocity. Control, *n* = 10 junctions in four embryos. $Ca^{2+}$ uncaging, *n* = 7 junctions in seven embryos.

Data information: Scale bars: 10 μm in (B, F).

AM (Fig 4C and D). The cross-sectional area of these control cells remained largely unaffected (Fig 4C and D). To establish a link between the increase in myosin II and the reduced cell area, we correlated both parameters with each other (Fig 4C and E). Indeed, we detect a strong correlation that the smaller the cell area is the higher the myosin II activity.

**Contracting cell induces cortical tension**

One expects that a contracting cell applies a force on the junctional complexes linking it to its neighbors within the epithelium (Fig 5A). To assess this action, we employed a reporter for tension across adherens junctions, based on the force-dependent conformational state of α-Catenin [27–30]. α-Catenin exhibits a force-dependent switch between two stable conformations. In the closed state, α-Catenin is bound to the Cadherin complex but does not bind to the D1 domain of Vinculin, because the central mechanosensitive modulatory (M) domain is inaccessible. In contrast, the central mechanosensitive modulatory (M) domain is exposed, when a force is applied to the molecule. α-Catenin bridges the Cadherin complex with the actin cytoskeleton and can thus sense and transduce forces acting on the adherens junctions. We thus introduced a GFP reporter based on the D1 domain of Vinculin (Fig EV4A) together with E-Cadherin-tomato inserted at the endogenous locus (Fig 5B and Movie EV5). We quantified the dynamics of VinD1-GFP fluorescence during an uncaging experiment (Fig EV4B). We detected a significant increase in the range of 10% of reporter fluorescence at the junctions next to the contracting target cell in the time scale of a minute. We did not detect such an increase at distant junctions, which served as a control in this experiment. As the time scale in response to uncaging by area change and VincD1 reporter fluorescence was comparable, we quantified their relationship and found a strong correlation between VincD1 reporter fluorescence and cell area (Fig EV4C).

The Vinc/E-cad ratio has been reported to correlate with junctional tension in *Drosophila* embryos [31]. We therefore quantified the dynamics of VincD1/E-cad fluorescence ratio in the CaLM-activated contracting cells (Fig 5B–E). We detect a 25% increase in VincD1/E-cad fluorescence ratio at the junctions next to the contracting target cell that appeared on a time minute scale. We did not detect such an increase at distant junctions, which served as a control in this experiment (Fig 5C). As the time scale in response to

uncaging by area change (Fig 5D) and VincD1/E-cad fluorescence ratio was comparable, we quantified their relationship. We plotted the mean of Vinc/E-cad ratio against the mean of cell area from six contracting cells and found a strong correlation between Vinc/E-cad ratio and cell area (Fig 5E). Furthermore, we assume that the CaLM-activated contracting cell applies a force to its neighbors within the epithelium. Following $Ca^{2+}$ uncaging, we therefore performed laser ablation on the first neighboring junctions of the CaLM-activated contracting cell (Fig 5F). The control ablation was performed on randomly selected junctions from the embryos injected with buffer (Fig 5F). We observed faster and greater recoil in $Ca^{2+}$ uncaging embryos compared within the control embryos (Fig 5F). The initial recoil velocity within 2-s after ablation is statistically significantly larger in $Ca^{2+}$ uncaging embryos than control embryos (Fig 5G). In summary, our experiments show that $Ca^{2+}$ uncaging induces cortical tension and CaLM-activated contracting cell applies a force on the junctional complexes linking it to its neighbors within the epithelium.

**Mechanism of $Ca^{2+}$-induced cell contraction**

Although we have observed that medio-apical myosin II accumulates in response to uncaging and that it correlates with the degree of cell contraction (Fig 4), the mechanism of how $Ca^{2+}$ induces contraction is unclear. At least two different mechanisms are conceivable. Firstly, $Ca^{2+}$ may activate myosin II activity via the generic pathway involving Rho kinase and phosphorylation of the regulatory light chain. Secondly, $Ca^{2+}$ may activate the myosin light-chain kinase or directly engage at the actomyosin filaments. We first tested whether the $Ca^{2+}$-induced contraction depended on Rho kinase by employing its specific inhibitor Y-27632 [32]. sqh-mCherry fluorescence is reduced obviously in Y-27632-injected embryos compared with water-injected embryos (Fig 6A and B). Following $Ca^{2+}$ uncaging, we did not detect any cell contraction in embryos treated with the Rho kinase inhibitor indicating that $Ca^{2+}$-induced contraction depends on Rho kinase (Fig 6E and I, EV4D, Movie EV7). $Ca^{2+}$ uncaging was functional in these embryos (Fig 6C and D, Movie EV6) as $Ca^{2+}$ fluorescence in Y-27632-treated embryos was comparable in timing and magnitude to wild-type embryos (Fig 6D). The dependence on Rho kinase strongly supports the model that the $Ca^{2+}$ signal acts via myosin II activation.

Rho kinase is activated by Rho signaling. RhoGEF2 is a major activator of Rho1 in the epidermal tissue during gastrulation, for example. We tested the dependence of the Ca$^{2+}$-induced cell contraction on RhoGEF2 by conducting the uncaging in embryos lacking RhoGEF2. The embryos from the female of *RhoGEF2* null mutation germline clones show multinucleated cell phenotype

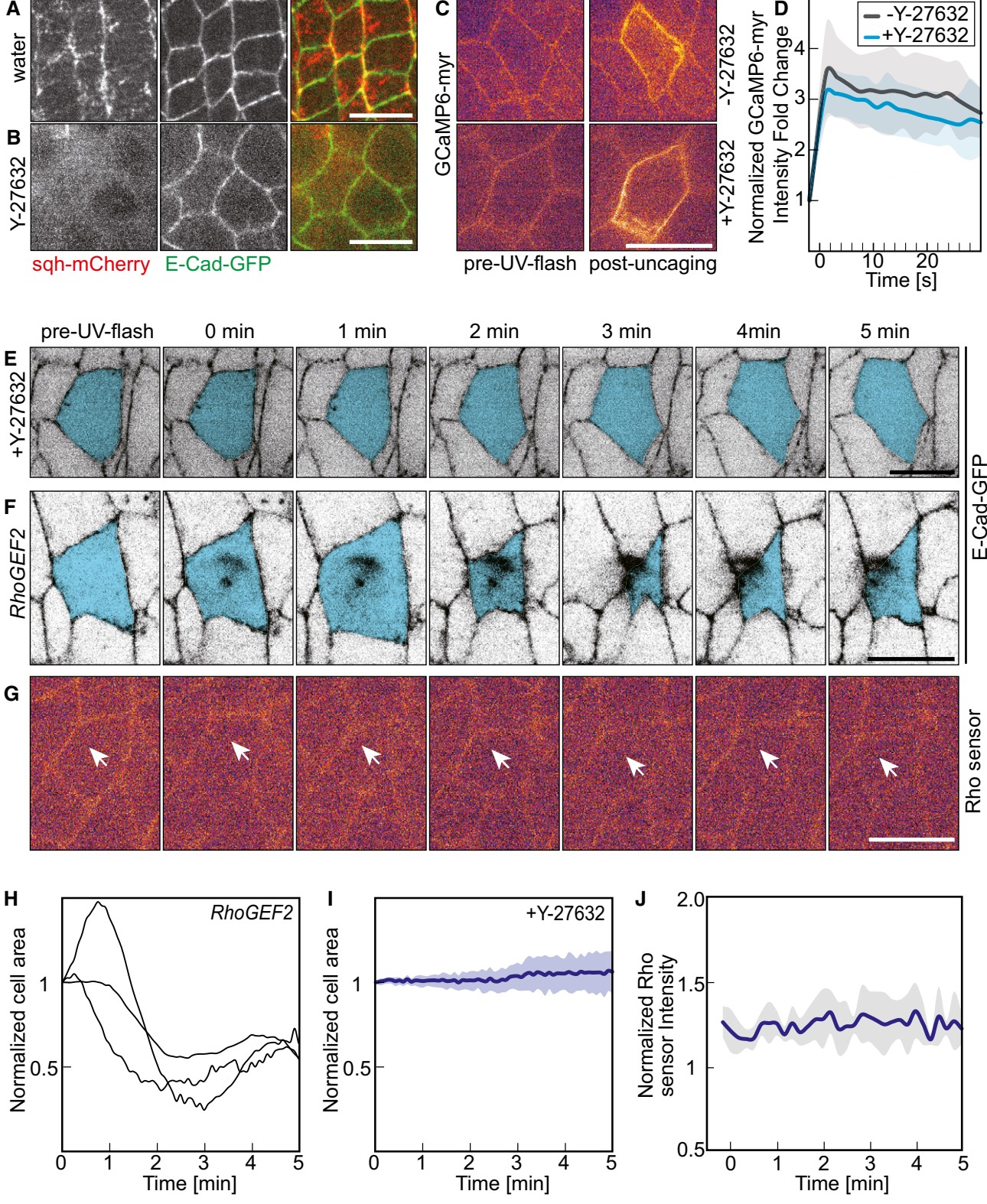

**Figure 6.**

**Figure 6. CaLM requires ROCK but not RhoGEF2.**

A, B   Confocal images of embryos expressing sqh-mCherry and E-Cadherin-GFP injected with Y-27632 (ROCK inhibitor, 10 mM) or water.
C, D   CaLM in embryos (stage 7, lateral epidermis) expressing a membrane-bound Ca²⁺ sensor (GCaMP6-myr) and injected with NP-EGTA, AM and Y-27632 as indicated. Images from time-lapse recording. (D) Fluorescence intensity of GCaMP-myr in the target cell with (black, the same data with Fig 1E) or without (blue) Y-27632. Mean (bold line) with standard deviation of the mean (ribbon band, six cells in six embryos).
E–G   Images from time-lapse recordings following CaLM (lateral epidermis, stage 7). Target cells marked in blue. (E) Co-injection of Rho kinase inhibitor Y-27632. (F) Embryos from *RhoGEF2* germline clones. (G) Embryo expressing a Rho sensor, and white arrows indicate the target cell.
H   Cross-sectional area traces of target cells normalized to initial size (the first frame of recording after uncaging) in embryos from *RhoGEF2* germline clone female following CaLM.
I   Cross-sectional area of target cells normalized to initial size (the first frame of recording after uncaging) in embryos injected with 10 mM Y-27632 (n = 8 cells in five embryos) following Ca²⁺ uncaging. Mean (bold line) with standard deviation of the mean (ribbon band).
J   Rho sensor fluorescence in target cells (n = 6 cells in six embryos) following Ca²⁺ uncaging. Mean (bold line) with standard deviation of the mean (ribbon band).

Data information: Scale bars: 10 μm in (A, C, E, F, G).

during cellularization as previous report (Fig EV4E) [33]. Quantification of the area dynamics of target cells revealed a behavior comparable in magnitude and timing to that in wild-type embryos (Fig 6F and H, Movie EV8). Lastly, we tested whether Rho1 was involved in mediating the $Ca^{2+}$ signal to Rho kinase by visualizing Rho1 activation with a sensor protein. The Rho sensor was functional, since we detected activation in cells undergoing cellularization (Fig EV4F) and cytokinesis (Fig EV4G). In contrast, we did not detect a change in Rho sensor fluorescence in response to $Ca^{2+}$ uncaging (Fig 6G and J, Movie EV9). In summary, we propose a mechanism linking $Ca^{2+}$ with myosin activation via Rho kinase but independent of Rho signaling via RhoGEF2.

## Discussion

We developed and validated a new method, which we designate CaLM to induce cell contraction in epithelial tissues with precise temporal and spatial control. The approach applies $Ca^{2+}$ uncaging, which has been well established in neurobiology, for example, to epithelial cell and developmental biology. By inducing $Ca^{2+}$ bursts in single or multiple cells, CaLM enabled us to induce contraction in selected cells to about half of the cross-sectional area within a minute. The induced contraction did not damage cells or perturb tissue integrity. To our best knowledge, this is the first report for optically controlled cell contraction on the minute scale and at single-cell resolution *in vivo* during epithelial tissue morphogenesis.

CaLM is based on UV laser-induced photolysis of a $Ca^{2+}$ chelator that has been widely employed [18]. The caged compound "NP-EGTA, AM" is membrane-permeant and thus allows convenient application on the tissue scale. The 355-nm pulsed UV laser, which we employ in this study, is compatible with modern objectives and can be conveniently mounted on standard live imaging microscopes via the epiport, for example. The dose of UV light depends on factors such as light scattering by the tissue and thickness of the sample. The actual dose of light at the target site can only be estimated and needs to be carefully titrated for the specific experimental system. We employed a genetically encoded $Ca^{2+}$ sensor protein for setting up the experimental conditions and testing the scale and time course of the $Ca^{2+}$ burst. Alternatively, $Ca^{2+}$ indicator dyes may be applied, depending on the sample. Besides the 355-nm pulsed

UV laser, we tested the suitability of a continuous wave laser at 405 nm, which is often installed at standard confocal microscopes. Using point scan illumination similar to FRAP protocols, we did not detect any increased signal of the GCaMP reporter (Fig EV5). The inefficiency of the 405-nm laser is consistent with the absence of significant absorbance of NP-EGTA at wavelengths longer than 400 nm [19]. Since our focus is to use CaLM to control contractility at single-cell resolution during tissue morphogenesis. In order to make the approach easy of handling, we only used 100× objective in all experiments. To stimulate contractility in multiple cells simultaneously, we applied CaLM in four amnioserosa cells (Fig EV3). Technically, CaLM should be applicable also to even more cells (e.g., 15–20 cells). Such experimental schemes will be tested in future investigations.

The detailed mechanism for the induced $Ca^{2+}$ burst and profile remains unclear. At this point, we do not know the origin and fate of $Ca^{2+}$ ions measured by the GCaMP sensor protein. A proportion of the $Ca^{2+}$ ions will be released from the photolyzed cage. It is conceivable, that in addition to this, intra- or extracellular $Ca^{2+}$ reservoirs are opened by $Ca^{2+}$-gated $Ca^{2+}$ channels, comparable to SERCA in muscle cells [34]. As the $Ca^{2+}$ levels return to low levels within minutes after uncaging, calcium ions may be exported from the cytoplasm to internal reservoirs such as ER or to the outside by $Ca^{2+}$ transporters.

The detailed mechanism of how $Ca^{2+}$ is functionally linked to contractile actomyosin also remains unclear, although there is no doubt that $Ca^{2+}$ is involved in regulation of contractility in many cell types [10–14,16]. It is clear that $Ca^{2+}$ does not directly act on actomyosin similar to the contractile system involving troponin C, given the time lag between $Ca^{2+}$ burst and contractility in the range of many seconds. The delayed response may indicate an indirect link via a signaling cascade.

In non-muscle cells, contractility is mediated by non-muscle myosin II, which is largely controlled by Rho-ROCK pathway [4]. In the cells we tested, we find that $Ca^{2+}$ is linked to this pathway at the position of ROCK. CaLM induces contractility by activating the medial pool of non-muscle myosin II, at least. Whether other pools of myosin II, such as junctional or basal myosin, are also activated remains unclear.

An expected consequence of a contracting cell within an epithelial tissue is a mechanical pull on its neighbors, which should be mediated by junctional complexes. This is an important issue, because an immediate application of CaLM is in tissue

morphogenesis with one of its central questions of how the temporal–spatial distribution of forces leads to changes in visible morphology. We tested the potential mechanical pull of target cells on its neighbors in two ways. Firstly, we applied a Vinculin-derived reporter, which preferentially binds to the open conformation of α-Catenin. α-Catenin undergoes a force-dependent conformational change, which opens a Vinculin binding site under mechanical pull [27–30]. Secondly, we directly assayed junctional tension in neighboring cells by measuring the initial recoil velocity after ablation. This experiment nicely shows the versatility of CaLM. The pulsed UV laser is employed for two tasks: firstly, the controlled uncaging in a single-target cell and secondly, shortly afterward the precise ablation of a single junction, all recorded in a movie of the tissue. CaLM will be, in principle, useful in many types of experiments concerning tissue morphogenesis. For example, intercellular coupling between neighboring cells poses a challenge to experimental design in studies of tissue morphogenesis. Here, cause and consequence cannot be easily distinguished without targeted activation of cellular contractility and precise external control of cellular behaviors. Thus, acute interference is mandatory for dissecting causal functional dependencies.

Taken together, CaLM allows us to control rapid cell contractility and generates forces within the tissue during morphogenesis. CaLM can be applied to a wide range of processes and organisms and should greatly improve our ability to study the causality of cell contractility in tissue mechanics and mechanotransduction *in vivo*. Importantly, CaLM does not require any genetically encoded protein and can be readily applied to any stock and genetic background. The independence from genetic constitution should vastly accelerate analysis and enable screening of mechanobiological cellular pathways and components, e.g., by comparing wide arrays of mutants to wild-type behavior. In addition, $Ca^{2+}$ uncaging is likely to open applications in manifold experimental systems with low genetic tractability. Importantly, UV-induced $Ca^{2+}$ uncaging leaves the entire visible spectrum available for optical interfacing with florescent protein indicators and opsin-based effectors. This in particular increases the options for simultaneously recording of cell and tissue behavior with the large palette of available fluorescent protein tags from CFP to RFP.

## Materials and Methods

### *Drosophila* strains and genetics

Fly stocks were obtained from the Bloomington Drosophila Stock Center, if not otherwise noted and genetic markers and annotations are described in FlyBase [35]. Following transgenes were used: UAS-GCaMP6-myr [21], E-Cadherin-GFP [36], E-Cadherin-mTomato [36], ubiquitin-E-Cadherin-GFP, Sqh-mCherry [26,37], UAS-GC3Ai, UAS-α-Catenin-TagRFP [23], Mat-Gal4-67,15 (D. St. Johnston, Cambridge/UK), and amnioserosa-Gal4 (Bloomington).

The allele $RhoGEF2^{04291}$ [33] together with $FRT^{2R, G13}$ was recombined with ubiquitin-E-Cadherin-GFP. *RhoGEF2* germline clones were generated and selected with $ovo^D$. First- and second-instar larvae were heat-shocked twice for 60 min at 37°C.

| *Drosophila* genotypes | Figures |
|---|---|
| *w; +/+; pUAS-GCaMP6-myr;* | Figs 1C, D, 6C, and EV5, Movies EV1 and EV6 |
| *w; ubiquitin-E-Cadherin::GFP; +/+;* | Figs 2B, C, F, and EV1D, E, Movies EV2 and EV3 |
| *w; E-Cadherin::GFP; +/+;* | Figs 2A, 3A–D, 5F, and EV2B, EV3, Movie EV4 and EV7 |
| *sqh^{AX3}; ubiquitin-E-Cadherin::GFP, Sqh::mCherry; +/+;* | Figs 4, and 6A and B |
| *w; pUAS-VinculinD1::GFP E-Cadherin::mTomato; +/+;* | Fig 5B, Movie EV5 |
| *w; ubiquitin-E-Cadherin::GFP RhoGEF2^{[04291]}, FRT^{[2R, G13]}; +/+* | Figs 6E and EV4E, Movie EV8 |
| *w; pUAS-α-Catenin::TagRFP; pUAS-GC3Ai;* | Fig EV2A |
| *w; Nanos-Anillin-RBD::tdTomato; +/+;* | Figs 6G, and EV4F and G, Movie EV9 |

### Cloning

VinculinD1 domain (aa6–257) (HindIII-Xho1) and eGFP (EcoR1-Xho1) were inserted between the EcoR1-Xho1 sites of a pUASt with attB sequence. PCR cloning was verified by sequencing of the fragments. pUASt-attB-VinculinD1-eGFP was inserted in chromosome II and recombined with E-Cad-mTomato. Homozygous lines were healthy and fertile.

The Rho sensor is a bicistronic cassette that contains tdTomato fused to the Rho-binding domain (RBD) from Anillin (aa748–1,239) followed by a P2A peptide and membrane marker, tdKatushka2, fused to the CAAX box from human KRAS. The utility of the Anillin-RBD for detecting regions of active Rho has been validated previously [38–40]. The Rho sensor was constructed by infusion cloning of three fragments into a Nanos cassette (Nanos promoter/5′utr and Nanos 3′utr) placed within P{valium22-(1)} tdTomato (Addgene—54653), (2) Anillin-RBD (DGRC-LD2793), and (3) p2a-tdKatushka2-caax (Addgene—56041). P2A and CAAX sequences were appended via primers. Transgenic lines were created by PhiC31 integrase-mediated transgenesis provided by BestGene at the following sites—attP2 and attP40. Homozygous lines were healthy and fertile.

### Embryo preparation and injections

Embryos were prepared as previously described [41]. Briefly, embryos (2–2.5 h at 25°C in Figs 1, 2, 4A–E and 5, and 15–17 h at 20°C in Figs 3 and 4F–K) were collected and dechorionated with 50% bleach (hypochloride) for 90 s, dried in a desiccation chamber for ~ 10 min, covered with halocarbon oil, and injected dorsally into the vitelline space in the dark at room temperature (~ 22°C). After injection, the embryos were incubated at room temperature in the dark for about 10 min prior to uncaging.

NP-EGTA, AM (Invitrogen) was prepared in 1× injection solution [180 mM NaCl, 10 mM HEPES, 5 mM KCl, 1 mM MgCl$_2$ (pH 7.2)] [11]. 2 mM NP-EGTA, AM was injected for $Ca^{2+}$ uncaging in epidermal cells, and 1 mM NP-EGTA, AM was injected for $Ca^{2+}$ uncaging in amnioserosa cells. To inhibit Rock activity, 10 mM Y-27632 (Sigma) in water was injected.

### Ca²⁺ uncaging and imaging

We employed a pulsed 355-nm YAG laser (DPSL-355/14, Rapp OptoElectronic) mounted on the epiport. We illuminated under the "Click and Fire" Mode on the "REO-SysCon-Zen" platform (Rapp OptoElectronic), while a movie was recorded via a spinning disk mounted on the side port (Zeiss ObserverZ1, 100×/oil, NA1.4, AxioCam MRm). For the images in Figs 2, 4, 5B, and EV2, EV3, the movies were recorded with an emCCD camera (Photometrics, Evolve 512) and the recording started about 20 s after Ca²⁺ uncaging. The intensity of the UV laser was adjusted so that no morphological changes were induced in 1× injection solution-injected embryos. The laser was applied for 1.5 s (around 300 pulses) per cell with 2.5% laser power (~ 0.5 mJ/cell).

The Ca²⁺ sensor GCaMP6-myr was maternally expressed with Mat-Gal4-67, 15 (Figs 1 and 6C). The cross-sectional images were recorded in GFP channel with a frame rate of 1/s. Ca²⁺ uncaging was applied during recording. Control experiments were conducted in embryos injected without NP-EGTA, AM but exposure to a similar UV laser pulse. To test Ca²⁺ uncaging with a 405-nm cw laser, the cross-sectional images were recorded in GFP channel with a frame rate of 0.2/s from the stage 7 embryo injected with NP-EGTA, AM and point scan illumination similar to FRAP bleaching was used for Ca²⁺ uncaging (Fig EV5).

E-Cad-GFP was the membrane marker for analysis of the cell dynamics after Ca²⁺ uncaging in epithelium. For the images in Figs 2, and EV2B and EV3, after uncaging, axial stacks of 3–4 images with 0.5 μm step size were recording in the GFP channel with frame rates of 0.2/s (Fig 2B–E) or 0.1/s (Figs 2F and G, and EV2B and EV3) with an emCCD camera (Photometrics, Evolve 512). The recording started about 20 s after Ca²⁺ uncaging. For the images in Figs 3, 6E, F, and EV1D, E, EV3A, B, the cross-sectional images were recorded in the GFP channel with a frame rate of 0.2/s. Ca²⁺ uncaging was applied during recording.

To analyze myosin dynamics after Ca²⁺ uncaging (Fig 4A and B), the GFP and mCherry channels were recorded simultaneously with a frame rate of 0.1/s for E-Cad-GFP and Sqh-mCherry. After uncaging, axial stacks of 3–4 images with 1 μm step size were recorded with an emCCD camera (Photometrics, Evolve 512). The recording started about 20 s after Ca²⁺ uncaging. Control experiments were conducted in embryos injected without NP-EGTA, AM but exposed with a comparable UV pulse.

VinculinD1-GFP was expressed under control of the AS-Gal4 driver in amnioserosa tissue. To analyze VinculinD1-GFP dynamics after Ca²⁺ uncaging (Fig 5A–E), GFP and mTomato channels were recorded simultaneously with a frame rate of 0.1/s with an emCCD camera (Photometrics, Evolve 512). The apical side of the amnioserosa tissue was acquired with four axial sections of 0.5 μm. The recording started about 20 s after Ca²⁺ uncaging.

α-Catenin-RFP and apoptosensor were expressed under control of the driver AS-Gal4 in the amnioserosa (Fig EV3A). Stage 14 embryos were collected and injected with 1 mM NP-EGTA. The Ca²⁺ uncaging was conducted in embryos expressing both α-Catenin-RFP and apoptosensor. After uncaging, axial stacks (10 images, 1 μm step size, GFP, and RFP channels) were recorded with a frame rate of 0.1/s on a spinning disk microscope (100×/oil, NA1.4) with an emCCD camera (Photometrics, Evolve 512). The recording started about 20 s after Ca²⁺ uncaging.

The Rho sensor was recorded in the GFP channel with a frame rate of 0.2/s (Fig 6G and J). Ca²⁺ uncaging was applied during recording. In Fig EV4F, axial stacks of 11 images with 0.5 μm step size were recording from an embryo undergoing cellularization with an emCCD camera (Photometrics, Evolve 512). In Fig EV4G, the cross-sectional images were recorded in the GFP channel with a frame rate of 0.2/s from a stage 8 embryo with an emCCD camera (Photometrics, Evolve 512).

In Fig 5A and B, embryos expressing sqh-mCherry and E-Cad-GFP were injected with water or 10 mM Y-27632, GFP, and mCherry channels were recorded simultaneously on a spinning disk microscope (Zeiss, 100×/oil, NA1.4) with an emCCD camera (Photometrics, Evolve 512). The apical planes of the embryo with four axial sections of 0.5 μm were acquired.

### Histology

Embryos were fixed, stained, and mounted as previously described [42]. Antibodies against the following antigens were used: Dlg (mouse, 0.4 μg/ml) [43], Arm (mouse M7A1, 0.4 μg/ml) [44], and Slam (rabbit, 1:5,000) [45]. Secondary antibodies were labeled with Alexa dyes (Invitrogen, 0.4 μg/ml). GFP booster labeled with ATTO488 (ChromoTek, 1:500) was used for E-Cad-GFP.

### Laser ablation

Stage 14 embryos expressing E-Cad-GFP were injected with 1 mM NP-EGTA, AM. Cross-sectional images were recorded in the GFP channel with a frame rate of 1/s from amnioserosa on a spinning disk microscope (100×/oil, NA1.4) with a CCD camera. Ca²⁺ uncaging was applied during recording. After the target cell started to contract, the 1ˢᵗ neighboring junction was ablated with the 10% of laser power, and 200 ms (around 40 pulses) exposure time during the recording mode (100× oil, NA 1.4) (Fig 5F). The control ablation was performed in the embryos injected with buffer without NP-EGTA, AM but exposed to the uncaging laser pulse. The junctions were selected randomly for ablation. The recoil velocity was calculated from the displacement of both ends of ablated junctions during the first 2 s.

### Image processing and analysis

The fluorescence intensity of GCaMP6-myr (Figs 1 and 6D) was measured manually with ImageJ/Fiji [46]. The integrated density (a.u.) was measured along the cell membrane and divided by the cell membrane length (μm) to get the mean fluorescence intensity $I_t$. The background $I_b$ was determined from the integrated density (a.u.), which was measured from the cytoplasm and divided by the measurement length (μm). The normalized GCaMP6-myr intensity fold increase was calculated as follows:

$$F/F_0 = (I_t - I_b)/(I_{-1} - I_{-1b})$$

where $I_t$ is the mean intensity at time $t$, $I_b$ is the mean intensity of the background at time $t$, $I_{-1}$ is the mean intensity at 1-s before UV illumination, and $I_{-1b}$ is the mean intensity of the background at 1 s before UV illumination.

To analyze cell dynamics after $Ca^{2+}$ uncaging, image stacks were projected by the "Max Intensity" option. The projected and cross-sectional images were segmented and tracked with "Tissue Analyzer" [47] in ImageJ/Fiji. Cell area measurements were carried out with ImageJ/Fiji. In Movie EV3, the Z-projected images were stabilized with "Image Stabilizer" [48].

To analyze myosin dynamics after $Ca^{2+}$ uncaging (Fig 4), the image stacks from sqh-mCherry embryos were projected with the "Max Intensity" option. Mean medio-apical Sqh-mCherry fluorescence intensity was measured manually with ImageJ and normalized with the initial fluorescence ($t = 0$).

To analyze Rho sensor dynamics, the fluorescence intensity of Rho sensor (Fig 6J) was measured manually with ImageJ/Fiji. The integrated intensity (a.u.) was measured along the cell membrane and divided by the cell membrane length (μm) to get the mean fluorescence intensity $I_t$. The background $I_{bt}$ represents the averaged fluorescence intensity (a.u.) within the cytoplasm. The normalized Rho sensor intensity was calculated as follows: $I = I_t/I_{bt}$.

The ratio of VincuinD1-GFP/E-cadherin-mTomato (Fig 5C) was generated by plugin "Ratio plus" in ImageJ/Fiji. The fluorescence intensity was measured along cell junctions and normalized to the initial fluorescence ($t = 0$). To analyze VinculinD1 and E-Cadherin dynamics, the fluorescence intensity of VinculinD1-GFP (in Fig EV4B) at cell junctions was measured manually with ImageJ/Fiji. The fluorescence intensity was measured along cell junctions and normalized to the initial fluorescence ($t = 0$).

**Expanded View** for this article is available online.

## Acknowledgements
We are grateful to Marion Silies, Stefan Luschnig, Adam Martin, Daniel St Johnston, and Magali Suzanne for materials. Stocks obtained from the Bloomington Drosophila Stock Center (NIH P40OD018537) were used in this study. BL is a New York Stem Cell Foundation—Druckenmiller Fellow. This work was in part supported by the Göttingen Centre for Molecular Biology (funds for equipment repair) and the Deutsche Forschungsgemeinschaft (DFG, FOR1756 GR1945/6-1/2, SFB937/TP10, and equipment grant INST1525/16-1 FUGG).

## Author contributions
DK conducted the experiments and analyzed the data. ZL generated the VinculinD1-GFP transgenic fly and analyzed the VinculinD1-GFP data. MH analyzed data and obtained in Figs 2E, 3E–G, 4E, and 5G. BL generated the Rho sensor transgenic fly. DK, FW, and JG conceived the study and wrote the manuscript. FW and JG supervised the study.

## Conflict of interest
The authors declare that they have no conflict of interest.

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
