## [Review Process File · EMBO Reports]

In vivo optochemical control of cell contractility at single cell resolution

Deqing Kong, Zhiyi Lv, Matthias Häring, Benjamin Lin, Fred Wolf, Jörg Großhans

Review timeline:

Submission date:	19 January 2019
Editorial Decision:	21 February 2019
Revision received:	20 July 2019
Editorial Decision:	28 August 2019
Revision received:	14 September 2019
Accepted:	2 October 2019

Editor: Deniz Senyilmaz-Tiebe

Transaction Report:

1st Editorial Decision

21 February 2019

Thank you for submitting your manuscript for consideration by EMBO Reports. It has now been seen by three referees whose comments are shown below.

As you can see, all referees express interest in the presented optochemical method to induce cell contractions *in vivo*. However, they also raise concerns that need to be addressed in full before we can consider publication of the manuscript here.

Given these constructive comments, I would like to invite you to revise your manuscript with the understanding that the referee must be fully addressed and their suggestions taken on board. Please address all referee concerns in a complete point-by-point response. Acceptance of the manuscript will depend on a positive outcome of a second round of review. It is EMBO Reports policy to allow a single round of revision only and acceptance or rejection of the manuscript will therefore depend on the completeness of your responses included in the next, final version of the manuscript.

REFeree REPORTS**Referee #1:**

This paper shows how uncaging EGTA with UV illumination in *Drosophila* embryos triggers calcium activation and subsequent contraction of the apical surface of the cell. Optogenetic methods have been recently developed in the *Drosophila* embryo to decrease or increase cell contractility and this new approach is interesting as it does not rely on genetic expression of optogenetic components and instead uses optochemical activation. Calcium activation is shown convincingly via use of the GCaMP6 reporter. Apical membrane contraction is linked to increased actomyosin contractility and an increased E-Cad/Vinculin ratio at junctions, suggesting that the apical actomyosin meshwork is pulling on E-cadherin complexes to shrink the cell surface. Increase of actomyosin contractility in response to EGTA uncaging requires Rok but not Rho or RhoGEF2. The experiments are reasonably

convincing although the number of cells uncaged tends to be on the low side. The manuscript feels preliminary at places and will need to be consolidated significantly before publication. In particular, the Discussion needs to be rewritten to focus on discussing the results.

1. In the discussion, the authors need to provide a more balanced assessment of how applicable their method is to other systems. In particular, could the authors comment on the following:

- the authors argue that their method doesn't interfere with imaging standard chromophores (GFP, Cherry)- but how specific to UV is uncaging of the compound e.g could uncaging occur at higher wavelengths than 355nm? Has this been tested?

- Although the uncaging method does not rely on genetic expression of components, the caged EGTA needs to be supplied to the tissue. Why have the authors used perivitelline injection of caged EGTA rather than the perhaps easier technique of injecting directly in the yolk of the syncytium embryo? To use this method in other tissues and model systems, how do the authors suggest the caged EGTA to be applied?

2- The authors uncage single cells within the embryonic tissues. What is the approximate size of the apical region of the cell illuminated by UV? The authors choose a membrane-tethered GCaMP6 reporter to report on calcium activation. The images show a confocal section of the cell at the level of adherens junctions but can they also see a calcium activation in the apical membrane? Presumably, only a subset of the caged EGDTA is uncaged upon a UV flash. Have the authors tried to repeat the UV pulse in the same cell to generate further contraction of the cell? Also, would the membrane permeant caged or uncaged reagent expected to diffuse from one cell to the next and if yes, at which rate?

3- In the germ-band tissue, some treated cells (5) stay contracted for the duration of the recording (~15 mins) while others (2) recover their initial area. Because n is low it is difficult to address if this is a real trend, but have the authors got an idea of why the cell deformation is sometimes reversible and sometime irreversible?

4- Have the authors tried to treat a patch or row of cells instead of single cells?

5- figure 2- the legend mentions 8 cells but there is only 7 traces in panel G

6- in one of the two tissues studied, the amnioserosa, the cells are naturally contracting overtime. Are the authors certain that they have examined enough cells to determine that the treated cells are contracting more quickly than the WT cells? In panel E of Figure 3, an averaged trace is mixed with single traces, which is confusing. With n being so low, it would be more rigorous to show the single traces for everything, or alternatively, to systematically show the single traces corresponding to all averaged curves in the supplementary figures.

7- Figure 4C: the Myosin II intensity is shown only for the treated cells: the WT control is missing. This data as it stands is very unconvincing.

8- Figure 4 E and K: presumably these graphs show the time evolution of the area and Myosin II intensity of a single treated cell and each data points are taken from each image of a time-lapse? This should be made clear in the legend. In E it would be clearer to have cell 1, 2, 3 rather than embryo 1, 2,3. In K, I am not sure that calculating a coefficient R is appropriate for a time evolution plot like this one.

9- about the Y-27632 treatment, is that also a perivitelline injection? Overall, the material and methods need more details (for example there is no reference for the RhoGEF2 allele; the type of microscope often is missing etc)

10- RhoGEF2 mutant: the text needs to mention that maternal and zygotic RhoGEF2 have been removed (by germ-line clones). What evidence do the authors have that the embryos they analysed are mutant for RhoGEF2 (did they check the phenotype etc)?

11- in the discussion, the authors need to comment about why Rock but not RhoGef2 or Rho seems to be required in the Ca^{++} -triggered contraction.

13 - are the black spots in Figure 5F damage in the overlying vitelline membrane from the UV flash? This suggests that significant energy is used- have the authors checked that the apoptotic program is not activated? GFP- reporters of apoptosis are available in *Drosophila* that could be used to test this.

14- last section of the results: the sentence about "Volume change " is unclear. Assuming that cell volume is not changed, the reduction in apical cell area should lead to the treated cell either adopting a wedge shape (like in apical constriction) or becoming more columnar. Have the authors looked on how the cell is changing shape in 3D?

15- What do the author mean when saying that there is no effect on neighbouring cells? Presumably when the apical cell area of the treated cell decreases, the area of the cell neighbor increases?

Referee #2:

In this manuscript, Kong et al report on a calcium chelator (o-nitrophenyl EGTA) which can be uncaged using UV irradiation and used as a tool to increase intracellular calcium concentration. This compound has been used previously to trigger exocytosis and neuronal transmitter release and study neuronal activity. Here the authors report on the use of this compound to study cell contractility during *Drosophila* embryogenesis. While I agree that it would be useful to use a non-genetically encoded chemical to induce cell contractility -especially in non-model organisms that are not amenable to genetic manipulation-, I am not convinced that the data shown in this manuscript demonstrate the efficacy and specificity of this compound to control cell contractility, and specifically apical constriction.

Major points:

-All experiments are performed in two tissues (stage 7 ectoderm and amnioserosa) that are already contractile and in which myosin is already apically localized and active. In Fig.2D only 5 control cells are analysed from 5 different embryos and only 8 activated cells in 8 different embryos? Essentially, on average only 1 cell/embryo. Cell in the lateral ectoderm are undergoing cycles of constriction and relaxation and at that stage also delaminate. How are the authors selecting which cells to quantify? Why do they consider only one cell/embryo? What is needed here is activation of multiple cells simultaneously (15/20 or more). If this tool is effective in inducing cell constriction, it should become obvious from this experiment that all the activated cells will constrict synchronously following the pattern of uncaging. The authors should perform this experiment also in tissues that are in a non-contractile state and where cells do not display apical myosin. This is particularly important as the effect on myosin activation demonstrated in Fig. 4A, D are impossible to distinguish from the normal behaviour of myosin in control cells. I am not convinced that we are seeing an upregulation of myosin activity here. In panel 4B, control cells (which are not selected as controls by the authors) show an equal up-regulation of myosin. And this is obvious given that apical myosin is active in this tissue. The correlation shown in panel 4E is meaningless as cells that have more myosin are also more likely to constrict. Probably the authors are just quantifying the normal behaviour of these cells. Indeed, it is not clear why increasing Ca concentration should induce constriction of the apical surface rather than whole-cell contractility.

-The 10% upregulation of vinculin/cadherin ratio shown in Figure 4G-I is very modest, not clear if statistically significant. If the authors want to demonstrate increase tension in activated cells, they should perform laser cut experiments and/or induce tissue level deformation by activating multiple cells at the same time.

Minor points

-In Figure 5 I am not sure I understand how do the authors conclude that the effects are RhoGEF-independent. Obviously, they must have used an hypomorphic allele otherwise embryos would not have developed to the stage they are looking at. Therefore, I do not think they can make this conclusion.

-In the abstract the effects are said to be reversible, in results (Fig 4G) they claim that in most of the cells is not reversible.

-Ca controls the activity of many proteins, therefore I question the specificity of this approach.

-Unclear how control cells were selected and why on average only 1 cell per embryo was analysed. In general, figure legends do not provide many details

-UV irradiation induces DNA damage, therefore apoptotic marker should be checked.

Referee #3:

Kong et al. used a cell-permeant Ca²⁺ photolabile chelator to induce single cell apical contraction. The paper is of general interest as the method should be widely applicable to study cell and tissue dynamics. I would recommend publication once the following points are addressed.

1. The following statement is incorrect or not supported by the authors data: "Most cells remained contracted during the following 15 min, whereas a minority of cells reexpanded to the original cross-sectional area (Fig. 2F, G)". In figure 2G, 2 cells out of 7 (around 28%) do not remain contracted.
2. "We did not observe that the exposure to UV laser and Ca²⁺ uncaging noticeably affected the further behaviour of the target cells and surrounding tissue. Cells showed typical oscillations with periods of a few minutes and amplitudes of 10-20% (Fig. 2F, G) [22,23]." The authors do not demonstrate that upon Ca²⁺ uncaging cells undergo normal dynamics at longer time scales. Also, in the amnioserosa the authors do not show that cells recover their original apical area following Ca²⁺ uncaging.
3. Why do the authors use different concentrations of NP-EGTA,AM for the experiments in the columnar cells versus the one in squamous epithelial cells?
4. Controls ("embryos injected with buffer") are missing in Figure 4. Notably and along this line, the following text was strikeout but not deleted "and a continuous decrease of fluorescence in control embryos, which may be due to bleaching".
5. The following statement is unclear: "These experiments rule out that Ca²⁺ induces a volume change independent of myosin II." Are the cells changing volume upon Ca²⁺ uncaging?
6. The authors conclude that apical contraction depends on Rok based on its inhibition by Y-26732. Yet, this inhibitor is not fully specific. The authors therefore need to use rok loss of function to further establish this point. Analysing Rok localisation could also reinforce its role in the Ca²⁺ response.
7. The authors establish that single cell contraction can be induced. Yet, to further show the relevance of the method, it would be essential that the authors test (i) whether large scale tissue deformation can be induced by Ca²⁺ uncaging in several neighbouring cells and (ii) whether repetitive Ca²⁺ uncaging leads to prolonged cell or tissue contraction and if this modifies tissue morphogenesis.

1st Revision - authors' response

20 July 2019

Referee #1:

This paper shows how uncaging EGTA with UV illumination in *Drosophila* embryos triggers calcium activation and subsequent contraction of the apical surface of the cell. Optogenetic methods have been recently developed in the *Drosophila* embryo to decrease or increase cell contractility and

this new approach is interesting as it does not rely on genetic expression of optogenetic components and instead uses optochemical activation. Calcium activation is shown convincingly via use of the GCaMP6 reporter. Apical membrane contraction is linked to increased actomyosin contractility and an increased E-Cad/Vinculin ratio at junctions, suggesting that the apical actomyosin meshwork is pulling on E-cadherin complexes to shrink the cell surface. Increase of actomyosin contractility in response to EGTA uncaging requires Rok but not Rho or RhoGEF2. The experiments are reasonably convincing although the number of cells uncaged tends to be on the low side. The manuscript feels preliminary at places and will need to be consolidated significantly before publication. In particular, the Discussion needs to be rewritten to focus on discussing the results.

We have conducted repetitions for most of the experiments to increase the sample sizes to five, at least. The sample sizes are specified in the figure legends. The discussion has been rewritten.

1. In the discussion, the authors need to provide a more balanced assessment of how applicable their method is to other systems. In particular, could the authors comment on the following:

- the authors argue that their method doesn't interfere with imaging standard chromophores (GFP, Cherry)- but how specific to UV is uncaging of the compound e.g could uncaging occur at higher wavelengths than 355nm? Has this been tested?

We tested uncaging with a continuous wave laser at 405 nm laser. This is a laser type installed at many standard confocal microscopes. Using point scan illumination similar to FRAP bleaching, we did not detect any increased signal of the GCaMP reporter. These data are shown as part of the supplemental data (Fig. S1). Our data are consistent with the absorbance spectrum of NP-EGTA (please see PNAS 91(1994) 187–191), which does not/very little absorb light at wave lengths longer than 400 nm.

- Although the uncaging method does not rely on genetic expression of components, the caged EGTA needs to be supplied to the tissue. Why have the authors used perivitelline injection of caged EGTA rather than the perhaps easier technique of injecting directly in the yolk of the syncytium embryo? To use this method in other tissues and model systems, how do the authors suggest the caged EGTA to be applied?

NP-EGTA is available in a form with linked fatty acid esters (NP-EGTA-am), which mediate membrane permeability and are cleaved by intracellular esterases. In this way NP-EGTA is trapped inside the cell. We inject NP-EGTA into the embryo/yolk for experiments with the early embryo. For the dorsal closure stage, we found injection into the extracellular perivitelline space more convenient. But we have not directly compared injection protocols. What is important is that NP-EGTA am is membrane permeable and does not need to be injected directly into cells. The NP-EGTA am compound has been used in neurobiology for many years in many different systems and is usually provided extracellularly. (Please see Graham C. R. Ellis-Davies, Chem. Rev (2008) 108, 1603–1613)

2- The authors uncage single cells within the embryonic tissues. What is the approximate size of the apical region of the cell illuminated by UV?

We illuminated a coated cover slide. Based on the scar on the slide, we estimated a diameter of 2–3 μm and thus an area of 5 μm^2 , which is slightly larger than the expected focal area of our optics.

The authors choose a membrane-tethered GCaMP6 reporter to report on calcium activation. The images show a confocal section of the cell at the level of adherens junctions but can they also see a calcium activation in the apical membrane?

This is a good point, which we have not yet managed to resolve. Unfortunately, UV illumination leaves a scar in the vitelline membrane which obscures measurements in this area. As the GCaMP signal is weak, a measurement at the lateral membrane, which extends along the axial axis of the focal volume, is easier than at the apical membrane. We expect a rapid diffusion of Ca^{2+} ions throughout the cell, since initial differences in the GCaMP signal at 2 s after uncaging are quickly lost. A few seconds later the GCaMP signal is uniform along the cell outline (please see Figure 1C). Presumably, only a subset of the caged EGDTA is uncaged upon a UV flash. Have the authors tried to repeat the UV pulse in the same cell to generate further contraction of the cell?

This is an interesting issue, which we have now experimentally tested. The data are shown in the supplemental data Figure S4. We exposed a selected cell in the amnioserosa three times with a UV pulse (0 min, 2.5 min, 5 min). We observed the typical contraction after the first pulse but no further obvious contractions after the second and third UV pulses (Fig. S4).

Also, would the membrane-permeant caged or uncaged reagent expected to diffuse from one cell to the next and if yes, at which rate?

As is done in the neurobiology field, we employ a membrane permeable version of NP-EGTA-am (acetomethyl-NP-EGTA), which contains fatty acid esters. After entry into cells the fatty acid moieties are cleaved off by intracellular esterases, which makes the cap membrane impermeable and thus traps it inside the cells. Although we have not tested this mechanism ourselves, what is important for our application is that the induced Ca^{2+} signal remains restricted to the target cell.

3- In the germ-band tissue, some treated cells (5) stay contracted for the duration of the recording (~15 mins) while others (2) recover their initial area. Because n is low it is difficult to address if this is a real trend, but have the authors got an idea of why the cell deformation is sometimes reversible and sometime irreversible?

This is an interesting issue for future experiment. It is conceivable that contracting cells could be prone to loose their oscillatory behavior, whereas relaxing cells may contract but reenter the oscillation state, for example. As testing this specific idea is beyond the scope of this study, we show here that both cases, reversible and irreversible contractions are observed. We conducted 3 uncaging experiments in amnioserosa cells with recording over 30 min. Two cells contracted irreversible, one cells relaxed after 10 min (Please see Fig. S6 B-C).

4- Have the authors tried to treat a patch or row of cells instead of single cells?

Following the referee's suggestion, we conducted such an experiment. We induced contraction by uncaging in a row of four cells in the amnioserosa (Fig. S5). An axial projection after five minutes shows a small groove in the tissue. Important for this study is that we demonstrate the induced contraction of a row of cells. Having the method in hand to induce cell contraction in a selected patch of cells will allow to test the contribution of contraction of morphogenetic movements such as furrow formation and invagination in future experiments.

5- figure 2- the legend mentions 8 cells but there is only 7 traces in panel G.

This was a mistake, which has now been corrected.

6- in one of the two tissues studied, the amnioserosa, the cells are naturally contracting overtime. Are the authors certain that they have examined enough cells to determine that the treated cells are contracting more quickly that the WT cells?

Following the recommendation, we increased the sample size to 7 cells and the control sample to 12 cells. Within the control the constriction rate averages out and remains constant over time (Fig. 3F), whereas a drop in averaged cross sectional area and increase of the averaged constriction rate are detected following uncaging (Fig. 3E). we also detect a statistically significant difference when comparing the maxima in the constriction rates (Fig. 3G).

In panel E of Figure 3, an averaged trace is mixed with single traces, which is confusing. With n being so low, it would be more rigorous to show the single traces for everything, or alternatively, to systematically show the single traces corresponding to all averaged curves in the supplementary figures.

As recommended we show now the average curve with a SEM band to simplify the presentation in Fig. 3E. In panel 3F we show the area traces of single cells, as the oscillatory behavior would not be visible in an averaged curve.

7- Figure 4C: the Myosin II intensity is shown only for the treated cells: the WT control is missing. This data as it stands is very unconvincing.

We improved this quantification as recommended showing the time course of myosin intensity in target cells next to the control cells (Figure 4D).

8- Figure 4 E and K: presumably these graphs show the time evolution of the area and Myosin II intensity of a single treated cell and each data points are taken from each image of a time-lapse? This should be made clear in the legend.

Yes, we now explicitly state this in the figure legend.

In E it would be clearer to have cell 1, 2, 3 rather than embryo 1, 2,3.

We increased the sample size from 3 to 6 and clearly state this in the figure legend.

In K, I am not sure that calculating a coefficient R is appropriate for a time evolution plot like this one. ?

With the panel Fig. 5E we would like to provide an argument for the suitability of the VincD1-GFP sensor protein in our system. Based on the assumption that cells with smaller cross-sectional areas are more contracted than cells with a larger cross-sectional area, the expectation is that larger forces

are present at smaller cells. We document the correlation of VincD1-GFP signal with cross sectional area. There is no time evolution involved.

9- about the Y-27632 treatment, is that also a perivitelline injection? Overall, the material and methods need more details (for example there is no reference for the RhoGEF2 allele; the type of microscope often is missing etc)

Yes, as Y-27632 is also membrane permeable, we injected into the perivitelline space. We extended the descriptions of methods of materials by adding more details.

10- RhoGEF2 mutant: the text needs to mention that maternal and zygotic RhoGEF2 have been removed (by germ-line clones).

The genetics for RhoGEF2 is now clearly referenced including our Development paper from 2005 (Grosshans et al, Development 2005). In fact, only the maternal genotype matters for the phenotype of embryos from RhoGEF2 germline clones, as no zygotic rescue is observed.

What evidence do the authors have that the embryos they analysed are mutant for RhoGEF2 (did they check the phenotype etc)?

Yes, we did this. This is easily done by the absence of larval hatching and the cuticle phenotype. Importantly, we also checked the characteristic phenotype of multinucleated cells during cellularization (Please see Fig. S8B).

11- in the discussion, the authors need to comment about why Rock but not RhoGEF2 or Rho seems to be required in the Ca^{++} -triggered contraction.

We address this issue in the discussion.

13 - are the black spots in Figure 5F damage in the overlying vitelline membrane from the UV flash?

Yes, this is the case. We have unfortunately not found a way to avoid this .

This suggests that significant energy is used- have the authors checked that the apoptotic program is not activated? GFP- reporters of apoptosis are available in Drosophila that could be used to test this. We employed a published reporter of apoptosis (Arnaud Ambrosini, *et al.*, Developmental Cell 50, 1–15, July 22, 2019) to rule out this possibility. We did the test in the amnioserosa, where we can demonstrate the functionality of the reporter due the normal presence of apoptotic cells during dorsal closure (Figure S6). We detected reporter signal in apoptotic cells but not in target cells subject to uncaging. We did Ca^{2+} uncaging and extended imaging period of 30 min, the target cell was present in the tissue without obvious morphological deviations (Fig S6B-C).

14- last section of the results: the sentence about "Volume change " is unclear. Assuming that cell volume is not changed, the reduction in apical cell area should lead to the treated cell either adopting a wedge shape (like in apical constriction) or becoming more columnar. Have the authors looked on how the cell is changing shape in 3D?

We fully agree to this comment. Given the technical difficulties in volumetric imaging, at least in our hands, we cannot make statements about cell volumes, and thus deleted the sentence about "Volume change ".

15- What do the author mean when saying that there is no effect on neighbouring cells? Thanks for asking for clarification. Certainly, a contracting cell influences its neighbors by pulling on them. Our statement refers to Ca^{2+} levels, i. e. that the uncaged Ca^{2+} remains restricted to the target cells. We corrected the text accordingly.

Presumably when the apical cell area of the treated cell decreases, the area of the cell neighbor increases?

In a tissue with very stiff cells, this would be expected due to area conservation. However, in soft tissues, the loss of area by the contracting cell may be compensated by expansion of cells close by as well as more distant cells. We have not conducted such measurements yet.

Referee #2:

In this manuscript, Kong et al report on a calcium chelator (o-nitrophenyl EGTA) which can be uncaged using UV irradiation and used as a tool to increase intracellular calcium concentration. This compound has been used previously to trigger exocytosis and neuronal transmitter release and study

neuronal activity. Here the authors report on the use of this compound to study cell contractility during *Drosophila* embryogenesis. While I agree that it would be useful to use a non-genetically encoded chemical to induce cell contractility -especially in non-model organisms that are not amenable to genetic manipulation-, I am not convinced that the data shown in this manuscript demonstrate the efficacy and specificity of this compound to control cell contractility, and specifically apical constriction.

Major points:

-All experiments are performed in two tissues (stage 7 ectoderm and amnioserosa) that are already contractile and in which myosin is already apically localized and active. In Fig.2D only 5 control cells are analysed from 5 different embryos and only 8 activated cells in 8 different embryos? Essentially, on average only 1 cell/embryo.

We have tried to avoid multiple uncaging experiments in the same embryo to have clearly defined conditions, as it is conceivable that Ca^{2+} release may have non-autonomous effects.

Cells in the lateral ectoderm are undergoing cycles of constriction and relaxation and at that stage also delaminate. How are the authors selecting which cells to quantify?

We conducted the experiments in the lateral epidermis at an early stage when no delamination is observed. We randomly selected cells for uncaging. Control cells without a UV pulse were selected as being most distant from the target cells.

Why do they consider only one cell/embryo?

In most of the experiments, we consider only one cell from one embryo, because we prefer to have very comparable starting conditions at stage 7. Give a period of about 20 to 30 min for one experiment, the second uncaging would be done already in stage 8. What is needed here is activation of multiple cells simultaneously (15/20 or more). If this tool is effective in inducing cell constriction, it should become obvious from this experiment that all the activated cells will constrict synchronously following the pattern of uncaging.

A central point of the study is that our tool can induce cell contraction at single-cell resolution. We fully agree to the referee that for other experiments synchronous contraction in multiple cells may be induced. To demonstrate the feasibility of the uncaging method we added Figure S5, where we present an experiment in which four cells in a row were subjected to uncaging (Please see Fig. S5). Consequently, all four cells contract and a small groove is observed.

The authors should perform this experiment also in tissues that are in a non-contractile state and where cells do not display apical myosin.

The cell type specificity of uncaging is an open and interesting question. To get a first step into this issue, we conducted Ca^{2+} uncaging in the head and dorsal region at stage 7, when these cells do not display apical myosin and do not display obvious changes in cross sectional area. Our results are presented in Figure S3. We detected cell contraction even in these cells. However, we do not like to draw strong conclusions, because we did not screen a series of cell types with characteristic contractile or non-contractile behavior or presence/absence of apical/junctional acto-myosin. In future experiments a comparison between opto-genetic and uncaging methods over such cell types may be interesting to reveal the lack of specific signaling pathways and lack of contractile capacity. This is particularly important as the effect on myosin activation demonstrated in Fig. 4A, D are impossible to distinguish from the normal behaviour of myosin in control cells. I am not convinced that we are seeing an upregulation of myosin activity here. In panel 4B, control cells (which are not selected as controls by the authors) show an equal up-regulation of myosin. And this is obvious given that apical myosin is active in this tissue. The correlation shown in panel 4E is meaningless as cells that have more myosin are also more likely to constrict. Probably the authors are just quantifying the normal behaviour of these cells.

The complication with the analysis of myosin upregulation is that fact that we get bleaching of myosin-Cherry during uncaging. For this reason, we compare cells exposed to a UV pulse in embryos injected with or without NP-EGTA. Such a comparison is now shown in Figure 4D. If a myosin were not affected by uncaging the ups and downs in Myosin intensity and cross sectional area would average out, as seen for the control cells. In the cells subjected to uncaging a stereotypic behavior is observed, however: a drop in cross sectional area (Fig. 4C) and an increase in Myo II (Fig. 4D). To demonstrate the correlation of myosin intensity and area change, we plotted the corresponding values for several target cells in Fig. 4E.

Indeed, it is not clear why increasing Ca concentration should induce constriction of the apical surface rather than whole-cell contractility.

This is an interesting question. At least for the two cell type which we have mostly investigated, lateral epidermis and amnioserosa, the behavior is not completely unexpected, as myosinII is clearly enriched apically and at the subapical junctions. If myoII were uniformly distributed over the apical, lateral and basal cortex, we would expect an overall increase in cell contractility, since we detected a uniform increase in GCaMP signal.

-The 10% upregulation of vinculin/cadherin ratio shown in Figure 4G-I is very modest, not clear if statistically significant. If the authors want to demonstrate increase tension in activated cells, they should perform laser-cut experiments and/or induce tissue level deformation by activating multiple cells at the same time.

We do not think that a 20% increase in Vinculin/ECad within 1–2 min is modest but rather very strong because it is in addition to the starting point of a tissue already under tension. The observed increase is consistent to other published reports (Girish R. Kale, NATURE COMMUNICATIONS, (2018) 9:5021). The increase is statistically significant (please see figure legend). As suggested to make the pulling on neighboring cells more obvious, we conducted ablation experiments for adjacent junctions and measured recoil velocities. The data are shown in Fig. 5F, G. In deed we revealed an averaged fourfold and statistically significant increased recoil velocity in adjacent junctions following uncaging. As requested, we conducted uncaging of multiple cells within a tissue. Contraction was induced in a row of four cells in amnioserosa, which resulted in a small groove after 5 min (Fig. S5). We did not expand this tissue scale experiments, since the central aim of the study is to present a method of non-invasive induction of cell contraction with single cell resolution.

Minor points

-In Figure 5 I am not sure I understand how do the authors conclude that the effects are RhoGEF-independent. Obviously, they must have used an hypomorphic allele otherwise embryos would not have developed to the stage they are looking at. Therefore, I do not think they can make this conclusion.

We and others previously reported the phenotype of embryos lacking RhoGEF2 (J. Großhans, *et al.*, Development 132 (2005) 1009-1020; Mojgan Padash Barmchi, *et al.*, JCB (2005) VOLUME 168 • NUMBER 4). The allele used behaved like strong alleles and molecular characterization strongly suggest that they represent strong alleles close to null alleles (strongly reduced or lacking mRNA, lack/strongly reduced immunostaining and western band in total embryonic extracts). These and other studies showed that RhoGEF2 is not essential for embryonic viability as such, as cuticles are formed. We describe the genetics of RhoGEF2 in the materials and methods and provide references.

-In the abstract the effects are said to be reversible, in results (Fig 4G) they claim that in most of the cells is not reversible.

We corrected this incorrect statement in the abstract.

-Ca controls the activity of many proteins, therefore I question the specificity of this approach.

We fully agree with this statement and we had the same doubts. For this reason, we tested the dependence on Rho kinase, RhoGEF2 and assays activation of Rho1. The dependence of cell contraction on Rho kinase shows that Ca uncaging induces a Rho kinase dependent process leading to cell contraction. Together with the increase in MyoII staining, we conclude that uncaging activates MyosinII. We do not rule out that other Ca²⁺ dependent processes are activated. A link of Ca²⁺ and myosin activation in non-muscle cells has been reported previously (L. He, *et al.*, 2010, Nat Cell Biol 12: 1133–1142; M. Suzuki, *et al.*, 2107, Development 144: 1307–1316).

-Unclear how control cells were selected and why on average only 1 cell per embryo was analysed. In general, figure legends do not provide many details

We have tried to avoid multiple uncaging experiments in the same embryo to have clearly defined conditions, as it is conceivable at Ca²⁺ release may have non-autonomous effects. In most of the experiments, we consider only one cell from one embryo, because we prefer to have very comparable starting conditions at stage7. Give a period of about 20 to 30 min for one experiment, the second uncaging would be done already in stage 8.

For the control experiments, we injected the embryos with buffer without NP-EGTA and subjected cells with a similar UV pulse. The control cells were selected randomly.

-UV irradiation induces DNA damage, therefore apoptotic marker should be checked. We employed a published live reporter of apoptosis (Arnaud Ambrosini, *et al.*, *Developmental Cell* 50, 1–15, July 22, 2019) to rule out this possibility. We did the test in the amnioserosa, where we can demonstrate the functionality of the reporter due the normal presence of apoptotic cells during dorsal closure (Figure S6). We detected reporter signal in apoptotic cells but not in target cells subject to uncaging. We did Ca²⁺ uncaging and extended imaging period of 30 min, the target cell was present in the tissue without obvious morphological deviations (Fig S6B-C).

Referee #3:

Kong et al. used a cell-permeant Ca²⁺ photolabile chelator to induce single-cell apical contraction. The paper is of general interest as the method should be widely applicable to study cell and tissue dynamics. I would recommend publication once the following points are addressed.

1. The following statement is incorrect or not supported by the authors data: "Most cells remained contracted during the following 15 min, whereas a minority of cells reexpanded to the original cross-sectional area (Fig. 2F, G)". In figure 2G, 2 cells out of 7 (around 28%) do not remain contracted.

We corrected this incorrect statement in the abstract.

2. "We did not observe that the exposure to UV laser and Ca²⁺ uncaging noticeably affected the further behaviour of the target cells and surrounding tissue. Cells showed typical oscillations with periods of a few minutes and amplitudes of 10-20% (Fig. 2F, G) [22,23]." The authors do not demonstrate that upon Ca²⁺ uncaging cells undergo normal dynamics at longer time scales.

We corrected the text.

Also, in the amnioserosa the authors do not show that cells recover their original apical area following Ca²⁺ uncaging.

We provide more data to clarify the issue of reversibility. We imaged three target cells in the amnioserosa for 30 minutes. One case was reversible, two cases remained contracted (Fig S6B-C).

3. Why do the authors use different concentrations of NP-EGTA,AM for the experiments in the columnar cells versus the one in squamous epithelial cells?

We titrated the concentration of NP-EGTA for each application protocol. Initially, we tried 2mM NP-EGTA, AM injection also in the case of the amnioserosa, which was unsuccessful.

Unexpectedly, lower concentrations at 1 mM and 0.5 mM were functional and lead to robust cell contractions after uncaging. At this point we can only speculate about the reasons for concentration dependence. One conceivable possibility is that specifically in the amnioserosa, a proportion of NP-EGTA remains unloaded and thus only sub-threshold amount of Ca²⁺ would be released.

4. Controls ("embryos injected with buffer") are missing in Figure 4.

The myosin intensity of control cells has been quantified and added in Figure 4D. Notably and along this line, the following text was strikeout but not deleted "and a continuous decrease of fluorescence in control embryos, which may be due to bleaching".

It has been corrected.

5. The following statement is unclear: "These experiments rule out that Ca²⁺ induces a volume change independent of myosin II." Are the cells changing volume upon Ca²⁺ uncaging?

Given the technical difficulties in volumetric imaging, at least in our hands, we cannot make statements about cell volumes, and thus deleted the sentence about "Volume change".

6. The authors conclude that apical contraction depends on Rok based on its inhibition by Y-26732. Yet, this inhibitor is not fully specific. The authors therefore need to use rok loss of function to further establish this point. Analysing Rok localisation could also reinforce its role in the Ca²⁺ response.

We are aware of this limitation and a geneticists we would prefer a clearly defined mutant situation such as with RhoGEF2, for example. Such a mutant situation is unfortunately not available for Rho kinase. Thus we stick to the state-of-the-art and employ the widely used inhibitor (S. L.Rogers, *et al.*, 2004, *Curr. Bio.*; C. Bertet, *et al.*, 2004, *Nature*; M. Bruno, *et al.*, 2009, *Nature cell biology*; Simoes Sde, M. *et al.*, 2010, *Dev. Cell*; J. Yu and R. Fernandez-Gonzalez, 2016, *elife*;). As

proposed by the referee, following the localization dynamics of Rok could be used as a supporting argument. Unfortunately, no functional GFP-tagged version of Rho kinase is available. Constructs which have been employed in previous papers have not been tested for complementation and are in the cases that we are aware of expressed in undefined amounts on top of the wild type allele. We have to await a version of Rok tagged at its locus or expressed from a genomic transgene in a Rok deficient background.

7. The authors establish that single cell contraction can be induced. Yet, to further show the relevance of the method, it would be essential that the authors test (i) whether large scale tissue deformation can be induced by Ca²⁺ uncaging in several neighbouring cells and (ii) whether repetitive Ca²⁺ uncaging leads to prolonged cell or tissue contraction and if this modifies tissue morphogenesis.

A central point of the study is that our tool can induce cell contraction at single-cell resolution. We fully agree to the referee that for other experiments synchronous contraction in multiple cells may be induced. To demonstrate the feasibility of the uncaging method we added Figure S5, where we present an experiment in which four cells in a row were subjected to uncaging (Fig. S5). Consequently, all four cells contract and a small groove is observed.

2nd Editorial Decision

28 August 2019

Thank you for submitting the revised version of your manuscript. It has now been seen by two of the original referees.

As you can see, both referees find that the study is significantly improved during revision and recommend publication. Before I can accept the manuscript, I need you to address some editorial points below:

- Please address the remaining concerns of the referees. I have discussed with referee #2, and we concluded that no further experiments would be necessary for publication, but please respond to all referee concerns in a point-by-point format and perform the necessary textual changes in the abstract and the manuscript text clarifying limitations of the technique as outlined by the referee.

REFeree REPORTS

Referee #1:

The authors have very significantly improved their manuscript, including increasing n for existing experiments and adding new experiments to demonstrate further the usefulness of their method. They have answered all my questions satisfactorily. This is now a very nice paper suitable for publication. To increase the impact and readability of the paper, I list below some suggestions that the authors might want to consider:

1- My questions are answered very clearly and compellingly in the rebuttal. At times however, these points are not yet as clear in the manuscript. The authors might want to check that clarifications made in the rebuttal find a place in the manuscript, for example:

- the fact that NP-EGTA enters the cell because it is membrane permeable but once in, cannot get out again because it is cleaved by intracellular esterases. This point should be emphasized in the introduction or the results: although it might be obvious to neurobiologists, this property of the compound might be unknown to most developmental cell biologists, the target audience for this paper.
- making clear in the results that they are looking at embryos null for RhoGEF2 and explaining briefly why + referring more explicitly to S8, which checks the expected phenotype.
- is Figure S3 described somewhere in the results?

2- In many fly papers, the material and methods contains a paragraph listing concisely the genotypes for each panel in figures. This helps greatly clarifying how experiments have been done and complement usefully the list of alleles and transgenes.

3- Typos:

line 122, 124: squamous

128 recording; neighbouring

133 rate

136 irreversibly

140 remove period

Fig. 3 E,F axes labels should be "normalized"

178 check sentence unclear

Fig S7 C normalized

201: sentence unclear- need to conclude what does S7C show.

Fig 5 E normalized

215 therefore

249 correct ref to Fig. 6G,J (not Fig 5)

263 caged

268 dose

298 " be" missing

Referee #2:

The authors have answered most of my concerns. One point that remains unclear is the extent to which this tool is applicable to stimulate contractility in cells that are not in a contractile state and the extent to which it can be used to stimulate contractility in multiple cells simultaneously (please see below). If no further data can be added to strengthen these two points, the authors should add a paragraph in the discussion explaining the current limitations of this tool given the experiments performed thus far. The abstract should be also revised accordingly as suggested below:

"The spatial and temporal dynamics of cell contractility plays a key role in tissue morphogenesis, wound healing and cancer invasion. Here we report a simple, single cell resolution, optochemical method to induce cell contractions in vivo during morphogenesis. We employed the photolabile Ca²⁺ chelator o-nitrophenyl EGTA to induce bursts of intracellular free Ca²⁺ by laser photolysis. Ca²⁺ bursts appear within seconds and are restricted to individual target cells. Cell contraction reliably followed within a minute, to about half of the cross-sectional area. Increased Ca²⁺ levels are reversible and the target cells further participated in tissue morphogenesis. Depending on Rho kinase (ROCK) activity but not RhoGEF2, cell contractions are paralleled with non-muscle myosin-II accumulation in the apico-medial cortex, indicating that Ca²⁺ bursts trigger non-muscle myosin II activation. Our approach can be, in principle, adapted to many experimental systems and species, as no specific genetic elements are required."

Specific points:

I asked to demonstrate contractility in 15-20 cells. In the new experiment presented in Fig.S5 only 4 cells are activated. In my opinion this implies this tool works only under certain conditions that have not been fully characterized. The authors argue their focus was to develop a tool to control contractility in single cells. However, this makes no sense in terms of biological outcome. If their tool works, it should work equally well to control contraction of a single- or of multiple cells at the same time, regardless of the experimentalist's motivation.

The authors argue in their response they have avoided " multiple uncaging experiments in the same embryo to have clearly defined conditions, as it is conceivable at Ca²⁺ release may have non-autonomous effects". How can this be reconciled with a tool that allows control of contractility in single cell? I think is dangerous to quantify the behavior of only one control/embryo. Given the variability of contractile behavior in the selected tissues, control cells should not be pre-selected. The contractile behavior of a single activated cell should be compared to the average behavior of all the cells (many) in the tissue.

The cell-type specificity of uncaging remains unclear. I asked to test Ca uncaging in non- contractile tissues. The results presented in Figure S3 show only one cell in the head and dorsal region of a stage 7 embryo. I do not find this experiment rigorous as only one cell is shown to contract. They

authors argue they do not like to draw "strong" conclusions about cell type specificity of uncaging and this should be explicitly discussed as suggested above.

The last point relates to the demonstration that Ca uncaging triggers myosin up-regulation. The authors do not directly show this point as they argue their myosin-mCherry probe bleaches upon uncaging. Why didn't they use myosin-GFP in combination with Cad-mCherry to directly demonstrate myosin upregulation?

2nd Revision - authors' response

14 September 2019

Referee #1:

The authors have very significantly improved their manuscript, including increasing n for existing experiments and adding new experiments to demonstrate further the usefulness of their method. They have answered all my questions satisfactorily. This is now a very nice paper suitable for publication. To increase the impact and readability of the paper, I list below some suggestions that the authors might want to consider:

We are grateful the Reviewer for his/her positive assessment of our work!

1- My questions are answered very clearly and compellingly in the rebuttal. At times however, these points are not yet as clear in the manuscript. The authors might want to check that clarifications made in the rebuttal find a place in the manuscript, for example:

-the fact that NP-EGTA enters the cell because it is membrane permeable but once in, cannot get out again because it is cleaved by intracellular esterases. This point should be emphasized in the introduction or the results: although it might be obvious to neurobiologists, this property of the compound might be unknown to most developmental cell biologists, the target audience for this paper.

We have added this point in the results.

- making clear in the results that they are looking at embryos null for RhoGEF2 and explaining briefly why + referring more explicitly to S8, which checks the expected phenotype.

Thank you for pointing it. We have added the explanations in the results.

- is Figure S3 described somewhere in the results?

Yes we have missed the description of Figure S3 was missed. In the revised version, Fig S3 is referred to in the main manuscript.

2- In many fly papers, the material and methods contains a paragraph listing concisely the genotypes for each panel in figures. This helps greatly clarifying how experiments have been done and complement usefully the list of alleles and transgenes.

Following this useful advice, we have added such a list in the material and methods.

3- Typos:

line 122, 124: squamous

128 recording; neighbouring

133 rate

136 irreversibly

140 remove period

Fig. 3 E,F axes labels should be "normalized"

178 check sentence unclear

Fig S7 C normalized

201: sentence unclear- need to conclude what does S7C show.

Fig 5 E normalized

215 therefore

249 correct ref to Fig. 6G,J (not Fig 5)

263 caged

268 dose

298 " be" missing

We incorporated the corrections in the revised version.

Referee #2:

We would like to thank the reviewer.

The authors have answered most of my concerns. One point that remains unclear is the extent to which this tool is applicable to stimulate contractility in cells that are not in a contractile state and the extent to which it can be used to stimulate contractility in multiple cells simultaneously (please see below). If no further data can be added to strengthen these two points, the authors should add a paragraph in the discussion explaining the current limitations of this tool given the experiments performed thus far.

We have added the explanations and clarification in the discussion as suggested.

The abstract should be also revised accordingly as suggested below:

"The spatial and temporal dynamics of cell contractility plays a key role in tissue morphogenesis, wound healing and cancer invasion. Here we report a simple, single cell resolution, optochemical method to induce cell contractions in vivo during morphogenesis. We employed the photolabile Ca²⁺ chelator o-nitrophenyl EGTA to induce bursts of intracellular free Ca²⁺ by laser photolysis. Ca²⁺ bursts appear within seconds and are restricted to individual target cells. Cell contraction reliably followed within a minute, to about half of the cross-sectional area. Increased Ca²⁺ levels are reversible and the target cells further participated in tissue morphogenesis. Depending on Rho kinase (ROCK) activity but not RhoGEF2, cell contractions are paralleled with non-muscle myosin-II accumulation in the apico-medial cortex, indicating that Ca²⁺ bursts trigger non-muscle myosin II activation. Our approach can be, in principle, adapted to many experimental systems and species, as no specific genetic elements are required."

The abstract has been revised as recommended.

Specific points:

I asked to demonstrate contractility in 15-20 cells. In the new experiment presented in Fig.S5 only 4 cells are activated. In my opinion this implies this tool works only under certain conditions that have not been fully characterized. The authors argue their focus was to develop a tool to control contractility in single cells. However, this makes no sense in terms of biological outcome. If their tool works, it should work equally well to control contraction of a single- or of multiple cells at the same time, regardless of the experimentalist's motivation.

The focus of our study was to develop a tool to spatially and temporally control cell contractility at single cell resolution during tissue morphogenesis. Based on the request of the reviewer we applied CaLM on groups of four cells within the amnioserosa (Fig EV4, previous Fig S5). The reason for this number is the technical set-up of our microscope. We have used a 100x objective in all experiments so far, which was needed for our applications (control of cell quadruplets during cell intercalation, mechanical cell coordination in the amnioserosa), but has a rather small field of view. We aim to have treated and untreated cells (in a distance from the treated cells) within the same images. We have not applied CaLM in settings with a 40x or even 25x objectives, yet. This would require a titration of the parameters. We expect that is possible, but we have simply not done this, because we have no suitable application for such settings.

The authors argue in their response they have avoided " multiple uncaging experiments in the same embryo to have clearly defined conditions, as it is conceivable at Ca²⁺ release may have non-autonomous effects". How can this be reconciled with a tool that allows control of contractility in single cell? I think is dangerous to quantify the behavior of only one control/embryo. Given the variability of contractile behavior in the selected tissues, control cells should not be pre-selected. The contractile behavior of a single activated cell should be compared to the average behavior of all the cells (many) in the tissue.

In case of multiple uncaging experiments within the same tissue, one would need to differentiate in the analysis whether a cell was subjected to uncaging as the first, second, third cell. With such a protocol a further variable would be introduced. In our view, the procedure is more reproducible and better defined, if only cells subject to uncaging as the first cells are compared. The developmental

stage of the embryos are clearly defined. In deed, it s fully consistent with the literature that epithelial cells are mechanically link and that mechanotransduction occurs in these tissues, i. e. that a contracting cell can influence the behavior of its neighbors. In cell quadruplets during germband extension we observe that contraction in old neighbors inhibited the contraction of the new neighbors. We observe this in a physiological situation as well as after CaLM. We do not preselect the control cells. The control cells are in distance to the target cell within the same images/movies. We calculate averages of these cells.

The cell-type specificity of uncaging remains unclear. I asked to test Ca uncaging in non- contractile tissues. The results presented in Figure S3 show only one cell in the head and dorsal region of a stage 7 embryo. I do not find this experiment rigorous as only one cell is shown to contract. They authors argue they do not like to draw "strong" conclusions about cell type specificity of uncaging and this should be explicitly discussed as suggested above.

It has been done.

The last point relates to the demonstration that Ca uncaging triggers myosin up-regulation. The authors do not directly show this point as they argue their myosin-mCherry probe bleaches upon uncaging. Why didn't they use myosin-GFP in combination with Cad-mCherry to directly demonstrate myosin upregulation?

The bleaching issue is unfortunately not easy to resolve, since most fluorescent proteins show a weak absorbance in the UV range (please see the spectra in www.fpbases.org, for example). Applying a very strong 355nm light pulse appears to be sufficient to bleach Myo-Cherry despite the low specific absorbance at this wave length. Swapping the tags will probably not resolve the issue. We observe less/no bleaching for E-Cad or for GCaMP likely because these are associated with the plasma membrane and thus outside of the focal volume.

3rd Editorial Decision

02 October 2019

Thank you for submitting your revised manuscript. I have now looked at everything and all looks fine. Therefore I am very pleased to accept your publication in EMBO Reports.

Corresponding Author Name: Deqing Kong

Journal Submitted to: EMBO reports

Manuscript Number: EMBOR-2019-47755